# Signal transduction in light-oxygen-voltage receptors lacking the active-site glutamine

Julia Dietler [1,8], Renate Gelfert[1,8], Jennifer Kaiser[1,8], Veniamin Borin [2], Christian Renzl[3], Sebastian Pilsl[3], Américo Tavares Ranzani[1], Andrés García de Fuentes[1], Tobias Gleichmann[4], Ralph P. Diensthuber[4], Michael Weyand[1], Günter Mayer [3,5], Igor Schapiro [2] & Andreas Möglich [1,4,6,7 ✉]

In nature as in biotechnology, light-oxygen-voltage photoreceptors perceive blue light to elicit spatiotemporally defined cellular responses. Photon absorption drives thioadduct formation between a conserved cysteine and the flavin chromophore. An equally conserved, proximal glutamine processes the resultant flavin protonation into downstream hydrogen-bond rearrangements. Here, we report that this glutamine, long deemed essential, is generally dispensable. In its absence, several light-oxygen-voltage receptors invariably retained productive, if often attenuated, signaling responses. Structures of a light-oxygen-voltage paradigm at around 1 Å resolution revealed highly similar light-induced conformational changes, irrespective of whether the glutamine is present. Naturally occurring, glutamine-deficient light-oxygen-voltage receptors likely serve as bona fide photoreceptors, as we showcase for a diguanylate cyclase. We propose that without the glutamine, water molecules transiently approach the chromophore and thus propagate flavin protonation downstream. Signaling without glutamine appears intrinsic to light-oxygen-voltage receptors, which pertains to biotechnological applications and suggests evolutionary descendance from redox-active flavoproteins.

[1] Department of Biochemistry, University of Bayreuth, 95447 Bayreuth, Germany. [2] Institute of Chemistry, The Hebrew University of Jerusalem, Jerusalem, Israel. [3] Life and Medical Sciences (LIMES), University of Bonn, 53121 Bonn, Germany. [4] Biophysical Chemistry, Humboldt-University Berlin, 10115 Berlin, Germany. [5] Center of Aptamer Research & Development, University of Bonn, 53121 Bonn, Germany. [6] Bayreuth Center for Biochemistry & Molecular Biology, Universität Bayreuth, 95447 Bayreuth, Germany. [7] North-Bavarian NMR Center, Universität Bayreuth, 95447 Bayreuth, Germany. [8] These authors contributed equally: Julia Dietler, Renate Gelfert, Jennifer Kaiser. ✉email: andreas.moeglich@uni-bayreuth.de

Light-oxygen-voltage (LOV) proteins form a sensory photo-receptor class that elicit a wide palette of physiological responses to blue light across archaea, bacteria, protists, fungi, and plants[1–3]. Complementing their eminent role in nature, LOV receptors also serve as genetically encoded actuators in optogenetics[4] for the spatiotemporally precise control by light of cellular state and processes[5]. At the heart of these responses lies the flavin-binding LOV photosensor module which belongs to the Per-ARNT-Sim superfamily[6] and comprises several α-helices (denoted Cα, Dα, Eα, and Fα) arranged around a five-stranded antiparallel β-sheet (strands Aβ, Bβ, Gβ, Hβ, and Iβ)[7,8] (Suppl. Fig. 1). Light absorption by the flavin triggers a well-studied photocycle[2,9–11], as part of which an initial electronically excited singlet state ($S_1$) decays within nanoseconds to a triplet state ($T_1$) (Fig. 1a). Likely via a radical-pair mechanism[12], $T_1$ reacts within microseconds to the signaling state, characterized by a covalent thioadduct between a highly conserved cysteine residue in the LOV photosensor and the C4a atom of the flavin isoalloxazine ring system. Once illumination ceases, the signaling state passively reverts to the resting state in the base-catalyzed dark-recovery reaction[13]. Thioadduct formation entails a hybridization change of the flavin C4a atom from $sp^2$ to $sp^3$ and concomitant protonation of the adjacent N5 atom. The resultant conversion of the N5 position from a hydrogen bond acceptor to a donor serves as the principal trigger[14] for a raft of conformational and dynamic transitions, that depending upon LOV receptor, culminate in order-disorder transitions[15], oligomerization[16], or other tertiary and quaternary structural changes[17]. A highly conserved glutamine residue in strand Iβ is situated immediately adjacent to the flavin and has been identified as instrumental in reading out the flavin N5 position and eliciting the downstream transitions. Supported by spectroscopy, structural and functional data, chemical reasoning, and molecular simulations[8,18–24], the glutamine

is widely held to rotate its amide sidechain to accommodate N5 protonation in the signaling state. As a corollary, additional hydrogen-bond rearrangements permeate the LOV photosensor and propagate towards the β-sheet scaffold. As recently proposed[25], glutamine reorientation, and signal propagation may be aided by transient rearrangements of two conserved asparagine residues that coordinate the pteridin portion of the flavin.

Notwithstanding the strong conservation of the glutamine residue and its established role in LOV receptors, recent reports indicate that at least in certain proteins, productive signaling responses to blue light may occur without the glutamine[26–28]. Potentially, these responses harness steric interactions rather than hydrogen-bonding changes as a means of signal transduction[26,29]. By contrast, reports on other LOV receptors considered the glutamine essential for eliciting blue-light responses[21,30].

To rationalize these conflicting findings and to provide further insight into signal transduction, here we systematically investigated the role of the conserved glutamine in several model LOV receptors (Fig. 1b and Suppl. Fig. 1). Unexpectedly, the glutamine residue is not essential in LOV signaling as productive blue-light responses were generally maintained even in its absence. Almost all other amino acids could functionally substitute for the conserved glutamine, with notable exceptions. High-resolution crystal structures of the paradigm *Avena sativa* phototropin 1 LOV2 (*As*LOV2) domain revealed that after glutamine substitution by leucine, closely similar structural changes are evoked by light as in the wild type. Based on structural data, chemical reasoning, and molecular simulations, we propose that in the absence of the glutamine, water molecules relay hydrogen-bonding signals from the flavin N5 position to the LOV β-sheet. The ability to transduce light signals without the glutamine appears to be an inherent, general trait of LOV receptors and may reflect their evolutionary origin. This notion finds support in the

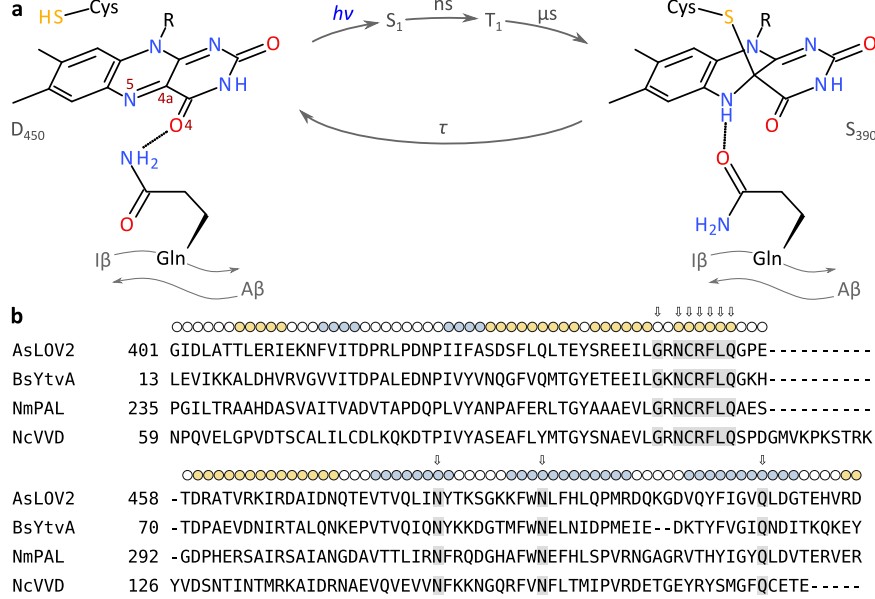

**Fig. 1 Photochemistry of light-oxygen-voltage receptors and sequences of proteins under study. a** Photocycle of light-oxygen-voltage (LOV) receptors. Absorption of blue light by the dark-adapted state ($D_{450}$) prompts the LOV receptor to traverse short-lived excited singlet ($S_1$) and triplet ($T_1$) states before assuming the light-adapted state ($S_{390}$), which is characterized by a thioadduct between the flavin atom C4a and the sidechain of a conserved cysteine. Adduct formation goes along with protonation of the N5 atom which entails changes in hydrogen bonding within the LOV receptor, particularly of a conserved glutamine residue situated in strand Iβ of an antiparallel β-pleated sheet. The light-adapted state passively decays to the dark-adapted state over a matter of seconds to hours, depending on the flavin surroundings. **b** Multiple sequence alignment of *A. sativa* phototropin 1 LOV2 (*As*LOV2)[15], *B. subtilis* YtvA LOV (*Bs*YtvA)[94], *N. multipartita* PAL LOV (*Nm*PAL)[46], and *N. crassa* Vivid LOV (*Nc*VVD)[59]. The secondary structure, as observed in *As*LOV2[47], is indicated on top, with α helices in tan and β strands in blue. Residues strongly conserved across LOV receptors[62] are highlighted by arrows and gray shading.

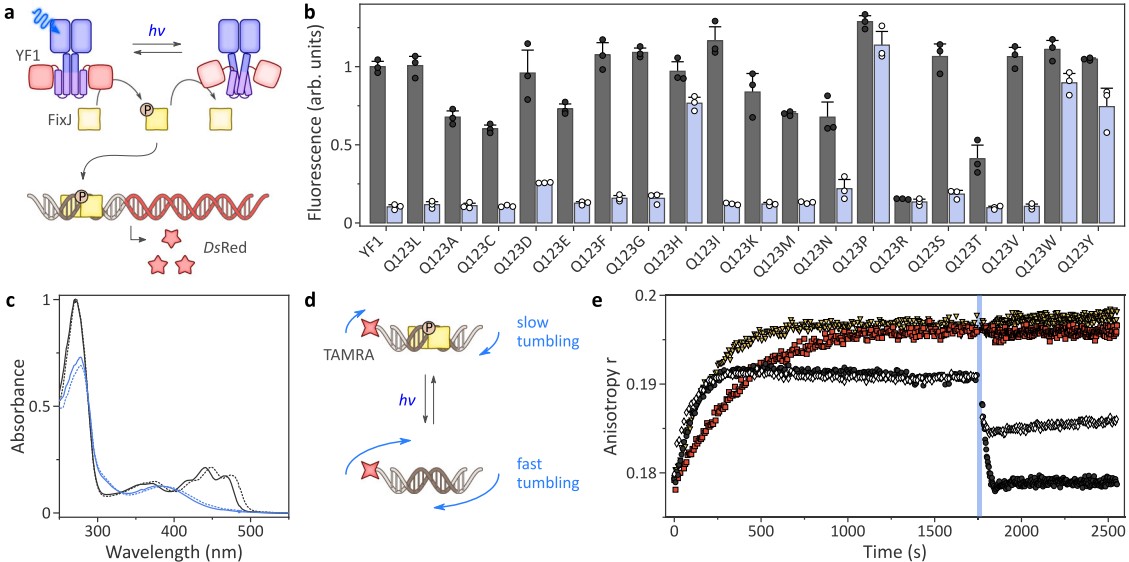

**Fig. 2 Activity and light response of YF1 variants. a** The net kinase activity of the variants was assessed in the pDusk-*Ds*Red setup[31], where alongside the response regulator *Bj*FixJ, YF1 drives the expression of the red-fluorescent reporter *Ds*Red in blue-light-repressed manner. **b** Normalized *Ds*Red fluorescence of *E. coli* cultures harboring pDusk plasmids encoding different YF1 variants. Cells were cultivated in darkness (black dots and gray bars) or under constant blue light (white dots and blue bars). Data represent mean ± s.d. of three biologically independent replicates. **c** Absorbance spectra of YF1 Q123L (solid lines) in its dark-adapted (black) and light-adapted states (blue), compared to the corresponding spectra of YF1 (dotted lines). **d** Schematic of the coupled fluorescence anisotropy assay to probe YF1 activity. Once phosphorylated in light-dependent manner (see **a**), *Bj*FixJ homodimerizes and binds to its cognate DNA operator sequence. Said operator is embedded in a TAMRA-labeled double-stranded DNA molecule, and *Bj*FixJ binding can be detected as an increase in fluorescence anisotropy due to decelerated rotational tumbling. **e** YF1 (black dots), YF1 Q123L (white diamonds), YF1 Q123H (yellow triangles), or YF1 Q123P (red squares) were incubated in darkness together with *Bj*FixJ and the TAMRA-labeled DNA. At time zero, the reaction was initiated by ATP addition and fluorescence anisotropy was recorded for 30 min. Samples were then illuminated for 30 s with blue light (blue bar), and the measurement continued. All experiments were repeated at least twice with similar results.

existence in nature of numerous LOV receptors that lack the conserved glutamine and presumably serve as blue-light receptors, as we confirm for a glutamine-deficient, proteobacterial LOV-diguanylate cyclase.

## Results

**Signal transduction in LOV receptors lacking the active-site glutamine**. To evaluate if and how LOV photosensors can transduce light signals to associated effector units in the absence of the conserved glutamine, we initially resorted to the histidine kinase YF1, as it allows the efficient assessment of signaling responses[31–33]. Together with the response regulator *Bj*FixJ, the engineered LOV receptor YF1 forms a light-sensitive two-component system (TCS) (Fig. 2a). *E. coli* cultures harboring the pDusk-*Ds*Red plasmid[31], which encodes the YF1/*Bj*FixJ TCS, exhibited strong expression of the red-fluorescent reporter *Ds*Red as YF1 acts as a net kinase in darkness[33]. Blue light converts YF1 to a net phosphatase, and accordingly, the *Ds*Red fluorescence decreased by around 12-fold (Fig. 2b).

To probe the role of the active-site glutamine (position Q123) in signal transduction, we substituted this residue for all 19 other canonical amino acids. Strikingly, most of the resultant glutamine-deficient YF1 variants prompted a blue-light-induced reduction of reporter gene fluorescence, similar to the original YF1 and almost regardless of which residue replaced the glutamine. These data clearly indicate that at least in the pDusk setup, the majority of residue substitutions, including alanine, cysteine, glutamic acid and leucine, leave light-dependent signal transduction largely unimpaired. Merely, the substitution by proline and the bulky aromatic amino acids His, Trp, and Tyr abolished responsiveness and resulted in high reporter expression independently of light. Similarly, the Q123R variant did not react

to light but exhibited constitutively low reporter fluorescence. The Q123A and Q123N exchanges were previously assessed in *Bacillus subtilis* YtvA from which YF1 derives[34,35]. As probed by photocalorimetry and in vivo analysis, the Q123A substitution slightly impaired signal transduction, but Q123N completely abolished any light responsiveness. Whereas the Q123A findings are consistent with the present data, YF1 Q123N retained attenuated light responses. The divergent observations for the Q123N exchange might be tied to the different effector modules in *Bs*YtvA and YF1. We note that asparagine in this position can principally support LOV signal transduction, as indicated by partial preservation of light responsiveness in the corresponding Q513N variant of *As*LOV2[21].

To glean additional insight, we expressed and purified the variants Q123H, Q123L, Q123P, and Q123R alongside YF1. Absorbance spectroscopy revealed flavin incorporation, as indicated by a three-pronged peak around 450 nm, for all variants but Q123R which failed to incorporate the chromophore and was prone to aggregation (Fig. 2c and Suppl. Fig. 2). As indicated by circular dichroism (CD) spectroscopy, the variants Q123H, Q123L, and Q123P were folded and adopted secondary and by inference, tertiary structure similar to YF1 (Suppl. Fig. 2). Upon blue-light exposure, YF1 and its variants Q123L and Q123P underwent the canonical LOV photocycle and adopted the thioadduct state with a characteristic absorption maximum near 390 nm (Fig. 2c and Suppl. Fig. 2). By contrast, the Q123H variant failed to form the adduct state despite incorporating the flavin cofactor, in line with earlier reports on *As*LOV2[36]. Only at high blue-light doses, the flavin absorption band slightly decreased in intensity but no band at 390 nm was formed. As reported earlier[20,21,36], replacement of the glutamine residue incurred a hypsochromic shift by around 8 nm of the flavin absorbance peak in both the dark-adapted and light-adapted

states. This spectral shift can tentatively be attributed to the loss of hydrogen bonding to the flavin O4 atom (see Fig. 1a) and is reminiscent of a bathochromic shift of similar magnitude during the photocycle of the so-called "sensors of blue light using flavin adenine dinucleotide" (BLUF)[37,38]. Taken together, the absorbance data account for the absent light responses in the pDusk context (see Fig. 2b) of the Q123H (no photocycle) and Q123R variants (no chromophore).

We next recorded the dark recovery after blue-light exposure and found the return to the dark-adapted state 10-fold decelerated in Q123L relative to YF1 (Suppl. Fig. 2). The Q123P variant exhibited even slower kinetics that was not completed even after several days. Given that the Q123L variant principally retained the capability of transducing signals (see Fig. 2b), we reasoned that modification of the active-glutamine provides an additional, little-tapped means of altering recovery kinetics[39] and thus modulating photosensitivity at photostationary state[40]. To explore this effect, we assessed the response of YF1 Q123L to pulsatile blue-light illumination[41] in the pDawn system that derives from pDusk but exhibits an inverted response to blue light[31]. The Q123L variant was toggled by much lower light doses than YF1, fully consistent with its retarded dark recovery (Suppl. Fig. 3). Compared to the V28I substitution, which also decelerates dark recovery by around 10-fold[39,41,42], the Q123L exchange was somewhat less sensitive to blue light. Combining the substitutions V28I and Q123L did not provide a further gain but slightly reduced the effective light sensitivity.

As the pDusk system only indirectly reports on the molecular activity of the receptors, we probed the catalytic activity and response to light of purified YF1 and its variants in a coupled fluorescence anisotropy assay (Fig. 2d). In darkness and in the presence of ATP, YF1 phosphorylates its cognate response regulator BjFixJ, thus prompting its homodimerization and binding of the FixK2 DNA operator sequence[14,43]. Phosphorylation-induced binding of BjFixJ to a short, double-stranded DNA molecule slows its rotational diffusion and causes an increase in fluorescence anisotropy of a 5′-attached tetramethylrhodamine (TAMRA) moiety. As noted above, blue light converts YF1 into a net phosphatase, thus promoting BjFixJ dephosphorylation, DNA dissociation, and a decrease in fluorescence anisotropy. Upon ATP addition, the dark-adapted YF1 and the Q123H, Q123L, and Q123P variants all exhibited increasing fluorescence anisotropy, albeit with somewhat differing kinetics and amplitude. Whereas Q123L showed a similar response as YF1, the Q123H and Q123P variants reached higher anisotropy values which likely reflects a higher degree of BjFixJ phosphorylation than the roughly 50% achieved for YF1[33]. The intrinsic equilibrium between the elementary histidine kinase and phosphatase activities of the TCS thus appears tilted towards the kinase state for Q123H and Q123P compared to YF1 and Q123L[44,45]. Upon blue-light application, the Q123L variant responded with a rapid fluorescence-anisotropy decay of around half the amplitude seen for YF1, indicating that light signals are transduced by this variant but less efficiently (Fig. 2e). Consistent with the pDusk reporter assay (see Fig. 2b), neither the Q123H nor the Q123P variant showed any response in their catalytic activities to blue light. In the case of Q123H, these observations are readily explained by its inability to undergo light-induced adduct formation and flavin N5 protonation. By contrast, the absorbance measurements unequivocally showed that Q123P can progress through the canonical LOV photocycle (see Suppl. Fig. 2). As the LOV photochemistry hence remains intact, signal transduction in the Q123P variant must be interrupted further downstream.

**LOV signal transduction can generally occur in the absence of the active-site glutamine**. We next addressed whether the

striking ability to transduce light signals without the conserved glutamine residue is specific for YF1 or more widely shared across LOV receptors. To this end, we examined light-dependent signaling responses in *Nakamurella multipartita* PAL[46], as a naturally occurring LOV receptor, and the *A. sativa* phototropin 1 LOV2 domain, as the arguably best-studied and optogenetically most widely used LOV module[5,15,47,48]. Notably, *Nm*PAL differs from YF1 by an unusual C-terminal arrangement of its LOV photosensor and binds a small RNA aptamer sequence-specifically and in a light-activated manner[46]. By embedding this aptamer directly upstream of the Shine-Dalgarno sequence in an mRNA encoding the fluorescent *Ds*Red protein, *Nm*PAL activity and response to light can be assessed in a bacterial reporter assay (Fig. 3a). In its dark-adapted state, wild-type *Nm*PAL has little affinity for the aptamer, and *Ds*Red is readily expressed. Light-induced binding by *Nm*PAL interferes with expression, presumably at the translational level, and reporter fluorescence is diminished by 10-fold (Fig. 3b). Using this assay, we tested the effect of replacing the active-site glutamine (residue Q347 in *Nm*PAL) with histidine, leucine, or proline. Consistent with the findings for YF1, the resultant Q347H and Q347P variants no longer exhibited light-induced changes in reporter fluorescence. As in the YF1 case, the proline variant had constitutive activity similar to the dark-adapted parental wild-type *Nm*PAL. Conversely, for Q347H we observed constitutively low fluorescence values, indicative of RNA binding and thus corresponding to light-adapted wild-type *Nm*PAL. This contrasts with YF1 where the corresponding histidine variant functionally corresponded to the dark-adapted state of the parental receptor. The Q347L variant exhibited a light-induced decrease of *Ds*Red fluorescence by around 17-fold, thus even surpassing the value for wild-type *Nm*PAL. Taken together, the results from the *Nm*PAL reporter assay are broadly consistent with the findings for YF1 in that the leucine substitution supported light responses to a significant extent whereas the histidine and proline substitutions incurred a loss of light-dependent signal transduction.

We next tested whether the ability of *Nm*PAL Q347L to transduce light signals extends to applications in eukaryotic cells. To this end, we harnessed an approach based on the translational repression of a luciferase reporter in HeLa cells[46] (Fig. 3c). Under blue light, wild-type *Nm*PAL can bind to an aptamer sequence embedded in the 5′-untranslated region of an mRNA and thereby represses luciferase expression by 10-fold relative to darkness (Fig. 3d). Upon introduction of the Q347L substitution into *Nm*PAL, blue-light-induced downregulation of reporter expression was maintained, albeit at reduced, fourfold efficiency.

To investigate photochemistry and RNA binding in detail, we expressed and purified *Nm*PAL wild-type and Q347L. In line with the reporter assays (see Fig. 3b–d), the Q347L variant retained flavin chromophore binding and underwent canonical LOV photochemistry upon blue-light exposure (Suppl. Fig. 4). As in YF1, replacement of the glutamine entailed a hypsochromic shift of the flavin absorption. Recovery kinetics after blue-light illumination were however only slowed down by 1.2-fold in the Q347L variant, rather than the 10-fold slowdown in YF1. Far-UV CD spectroscopy showed that *Nm*PAL and its Q347L variant adopt closely similar secondary structures (Suppl. Fig. 4c). We next assessed the binding of *Nm*PAL wild-type and Q347L to a TAMRA-labeled RNA aptamer by fluorescence anisotropy[46] (Fig. 3e, f). Wild-type *Nm*PAL bound the RNA with an affinity of $(45.4 \pm 5.4)$ nM in its light-adapted state but showed much-reduced interaction in darkness $[(1200 \pm 93)$ nM]. Under the same conditions, *Nm*PAL Q347L interacted with the aptamer somewhat less strongly under blue light $[(202.5 \pm 8.5)$ nM] but exhibited more pronounced residual binding in darkness with an affinity of around $(930 \pm 70)$ nM. Thus, light-dependent signal

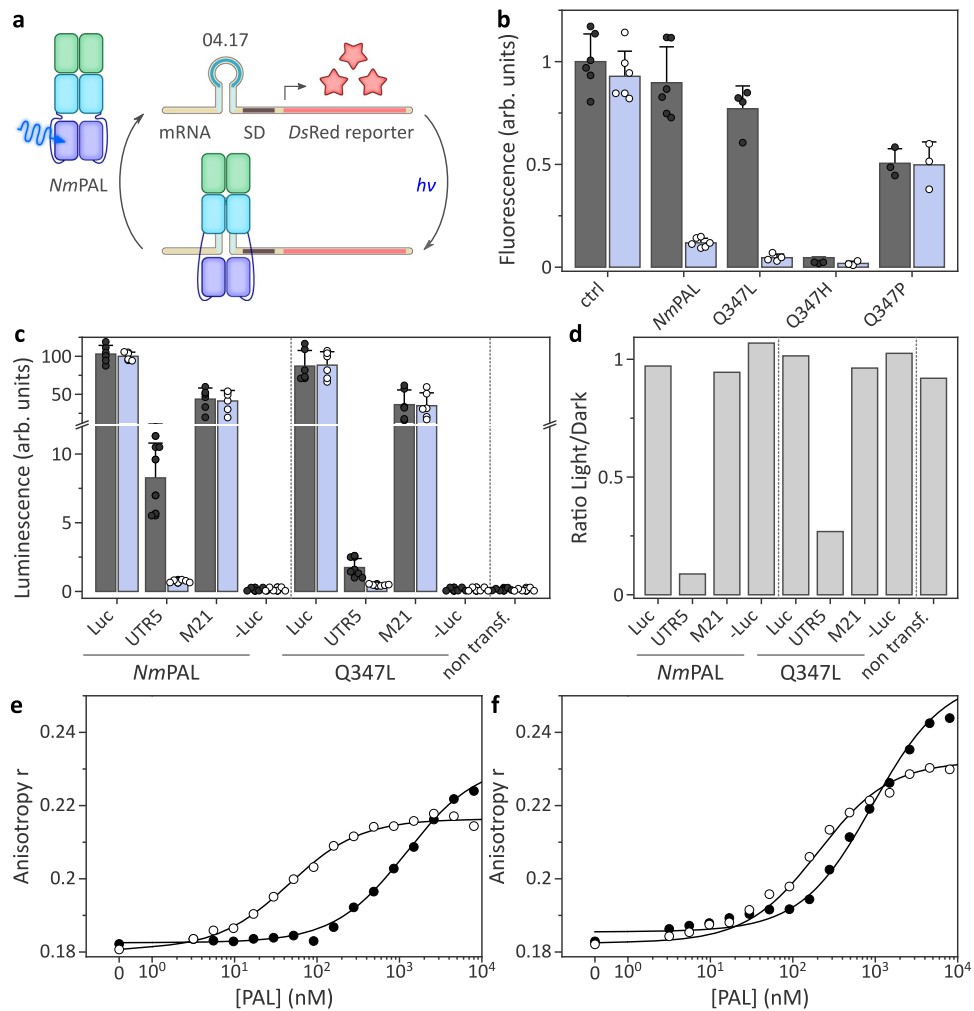

**Fig. 3 Activity and light response of *Nm*PAL variants. a** By embedding a specific aptamer (denoted 04.17) near the Shine-Dalgarno sequence (SD) of an mRNA encoding *Ds*Red, the expression of the fluorescent reporter can be modulated with *Nm*PAL as a function of blue light[46]. In darkness, *Nm*PAL shows little affinity for the aptamer, and expression ensues. Under blue light, *Nm*PAL binds and thus attenuates expression. **b** *E. coli* cultures harboring different *Nm*PAL variants and the reporter system depicted in panel a were cultivated in darkness (black dots and gray bars) or under blue light (white dots and blue bars). Normalized *Ds*Red fluorescence represents mean ± s.d. of at least three biologically independent samples. **c** *Nm*PAL variants were expressed in HeLa cells to translationally repress expression of a luciferase reporter, conceptually similar to the setup shown in panel a but using the 53.19 aptamer[46]. Bars represent the mean ± s.d. of luminescence acquired for six biologically independent samples incubated in darkness (black dots and gray bars) or under blue light (white dots and blue bars). UTR5 refers to the intact reporter system giving rise to *Nm*PAL-mediated light responses;[46] in M21, *Nm*PAL binding is disrupted by a mutation in the target aptamer, and light responsiveness is abolished. As positive and negative controls, luciferase was constitutively expressed (Luc) or left out altogether (-Luc). **d** Ratio of the luminescence values obtained under light and dark conditions. **e** The interaction of wild-type *Nm*PAL with the TAMRA-labeled 04.17 aptamer was assessed in its dark-adapted (black dots) and light-adapted states (white dots) by fluorescence anisotropy[46]. The line represents a fit to a single-site binding isotherm. **f** As in **e** but for *Nm*PAL Q347L. Experiments in panels e and f were repeated twice with similar results.

transduction is principally retained in *Nm*PAL Q347L but is impaired compared to the wild-type receptor, similar to the observations on YF1.

We next turned to the LOV2 domain from *A. sativa* phototropin 1 (*As*LOV2) as a widely studied paradigm[15,36,47,49,50] that underpins manifold applications in optogenetics[5,51,52]. Whereas *As*LOV2 wild-type, Q513H, and Q513L could all be produced with good yield and purity, the Q513P variant suffered from poor expression and severe aggregation, thus precluding its further analysis. The Q513H and Q513L variants incorporated flavin cofactors and exhibited a hypsochromically shifted absorbance spectrum compared to wild-type *As*LOV2 (Suppl. Fig. 5), as seen for YF1 and *Nm*PAL. Under blue light, the Q513L variant populated the thioadduct state which recovered to the resting state in darkness with kinetics around 22-fold slower than

those of the wild-type domain (Suppl. Fig. 5). By contrast, the Q513H variant failed to undergo the canonical LOV photochemistry, consistent with the YF1 and *Nm*PAL scenarios. The dark-adapted wild-type, Q513H, and Q513L proteins showed closely similar far-UV CD spectra, characterized by two minima of the molar ellipticity per residue, $[\Theta]_{MRW}$, at ~208 nm and 220 nm, and consistent with the mixed αβ fold of *As*LOV2[47] (Fig. 4 and Suppl. Fig. 5). Exposure to blue light diminished the amplitude of the minima by ~30–35% for both *As*LOV2 wild-type and Q513L, reflecting the unfolding of the N-terminal A'α and the C-terminal Jα helices[50]. However, given the relatively fast recovery of *As*LOV2 wild-type (see Suppl. Fig. 5), a significant return to the dark-adapted state is expected during the spectral scan (taking ~1 min). We hence monitored the α-helical CD signal at (208 ± 5) nm immediately after the withdrawal of blue

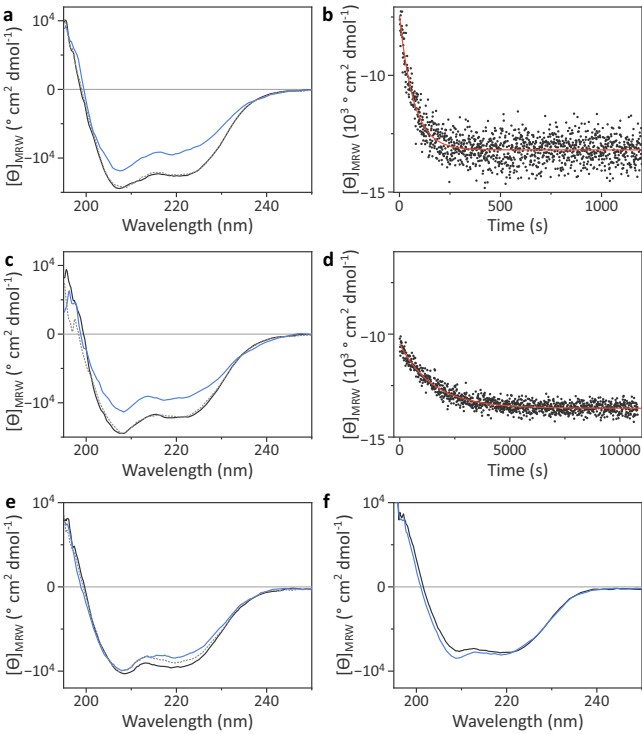

**Fig. 4 Light response of AsLOV2 variants. a** Far-UV circular dichroism (CD) spectra of AsLOV2 in its dark-adapted (black) and light-adapted states (blue), and after dark recovery (gray dotted). **b** Recovery reaction of AsLOV2 following blue-light exposure, as monitored by the CD signal at $(220 \pm 5)$ nm. Data were fitted to a single-exponential decay (red line), yielding a recovery rate constant $k_{-1}$ of $(1.43 \pm 0.05) \times 10^{-2}\,\mathrm{s}^{-1}$. **c** As **a** but for AsLOV2 Q513L. **d** As panel **b** but for AsLOV2 Q513L, with $k_{-1}$ amounting to $(6.61 \pm 0.15) \times 10^{-4}\,\mathrm{s}^{-1}$. **e** As **a** but for AsLOV2 C450A:Q513D. **f** As **a** but for AsLOV2 C450A:Q513D ΔA′α ΔJα. Experiments were repeated at least twice with similar results.

light (Fig. 4b, d). The kinetic measurements revealed that the initial amplitude of the light-induced CD change in AsLOV2 Q513L was only half that in the wild-type protein. For both variants, the CD spectra fully recovered to their original states (Fig. 4) with kinetics matching those of the photochemical recovery probed by absorbance measurements (see above and Suppl. Fig. 5). In agreement with our findings, an earlier study reported light-induced CD changes for the Q513L variant but at much-reduced amplitude compared to wild-type AsLOV2[21]. Taken together, our CD measurements suggest that glutamine replacement by leucine (but not by histidine) qualitatively, if not quantitatively, preserves light-induced signaling responses, fully consistent with the results on the other LOV receptors.

A previous investigation showed that LOV receptors can trigger productive signaling responses even when devoid of their active-site cysteine[14]. Blue light then promotes photoreduction of the flavin chromophore from its oxidized quinone form to the partially reduced neutral semiquinone (NSQ), which shares with the thioadduct state a protonated N5 atom and is thus capable of intact signal transduction[14]. We consequently wondered whether these aspects also hold true for LOV receptors that lack both the conserved cysteine and glutamine residues. As cysteine-deficient LOV receptors can efficiently sensitize molecular oxygen[53], pertinent experiments may be complicated by reactive oxygen species (ROS), which potentially disrupt or obscure genuine signaling responses to blue light. We, therefore, opted to assess the effect of combined cysteine and glutamine removal by CD spectroscopy in the isolated AsLOV2 module, as a comparatively

well-defined and tractable experimental setup. Replacement of the active-site cysteine (residue 450) in wild-type AsLOV2 by alanine abolished canonical photochemistry but the NSQ yield was poor, even at prolonged illumination and in the presence of the reductant TCEP (Suppl. Fig. 5). Nor did the additional introduction of the Q513L exchange significantly enhance NSQ formation. We thus capitalized on the recent finding that replacement of the active-site glutamine by aspartate in an *Arabidopsis thaliana* phototropin LOV domain greatly promoted photoreduction to the NSQ[54]. Given that the corresponding Q123D substitution in YF1 retained signaling capability (see Fig. 2b), we generated AsLOV2 Q513D and the doubly substituted C450A:Q513D variant. Absorbance spectroscopy revealed that the Q513D variant underwent the canonical LOV photochemistry and formed the thioadduct state (Suppl. Fig. 5). CD spectroscopy showed a light-induced 25% loss of $[\Theta]_{\mathrm{MRW}}$, indicating that the Q513D variant can indeed transduce blue-light signals (Suppl. Fig. 5). In the case of AsLOV2 C450A:Q513D, blue light drove rapid conversion to the NSQ state even without the addition of reductants, as determined by absorbance spectroscopy (Suppl. Fig. 5). Analysis by CD spectroscopy identified an ~10% loss in α-helical content upon blue-light exposure (Fig. 4e). Notably, the underlying conformational change was reversible, and upon slow reoxidation of the NSQ to the quinone state, the CD signal recovered over time. To ascertain that the change in helical content truly involves the A′α and Jα helices as in wild-type AsLOV2, we generated the AsLOV2 C450A:Q513D ΔA′α ΔJα derivative with these helices truncated. Consistent with the removal of A′α and Jα, this variant exhibited a 20% reduction in $[\Theta]_{\mathrm{MRW}}$ at 220 nm (Fig. 4f). Rather than a decrease, blue light elicited a small signal gain at ~208 nm whose molecular origin is unclear. Given that both the quinone and NSQ states strongly absorb in the far-UV region, we tentatively ascribe this signal to flavin photoreduction. As we observed no loss in α-helical structure, the light-induced structural changes in AsLOV2 C450A:Q513D are likely caused by the partial unfolding of the terminal A′α and Jα helices. Although the amplitude of the structural response is greatly reduced compared to wild type, it is striking that light-induced responses can be elicited in the absence of two highly conserved active-site residues.

**Molecular bases of LOV signal transduction without the active-site glutamine.** The above findings compellingly show that several LOV receptors transduce light signals in the absence of the active-site glutamine, long considered essential. To arrive at a molecular understanding, we solved the crystal structures of AsLOV2 wild-type and Q513L in the dark-adapted states to resolutions of 1.00 Å and 0.90 Å, respectively. Notably, both AsLOV2 variants formed crystals at the previously published solution conditions[47] and adopted the same space group with closely similar cell dimensions (Suppl. Tables 1 and 2). To additionally acquire information on the light-adapted state, we pursued a freeze-trapping strategy. Dark-grown crystals were exposed to blue light and rapidly cryo-cooled, X-ray diffraction was recorded, and structures were refined to resolutions of 1.09 Å (wild type) and 0.98 Å (Q513L) (Suppl. Tables 1 and 2). Although the crystal lattice stands to influence any structural rearrangements, in the past light-induced conformational transitions could thus be resolved for several LOV receptors[8,19,47,55], if likely at reduced amplitude and extent than in solution. Overall, the dark- and light-adapted states of AsLOV2 wild-type and Q513L exhibited closely similar structures with pairwise root-mean-square displacement (rmsd) values of 0.33–0.36 Å for the main-chain atoms of residues 404–546 (Suppl. Fig. 6). Differences among the four structures were subtle and concentrated on the

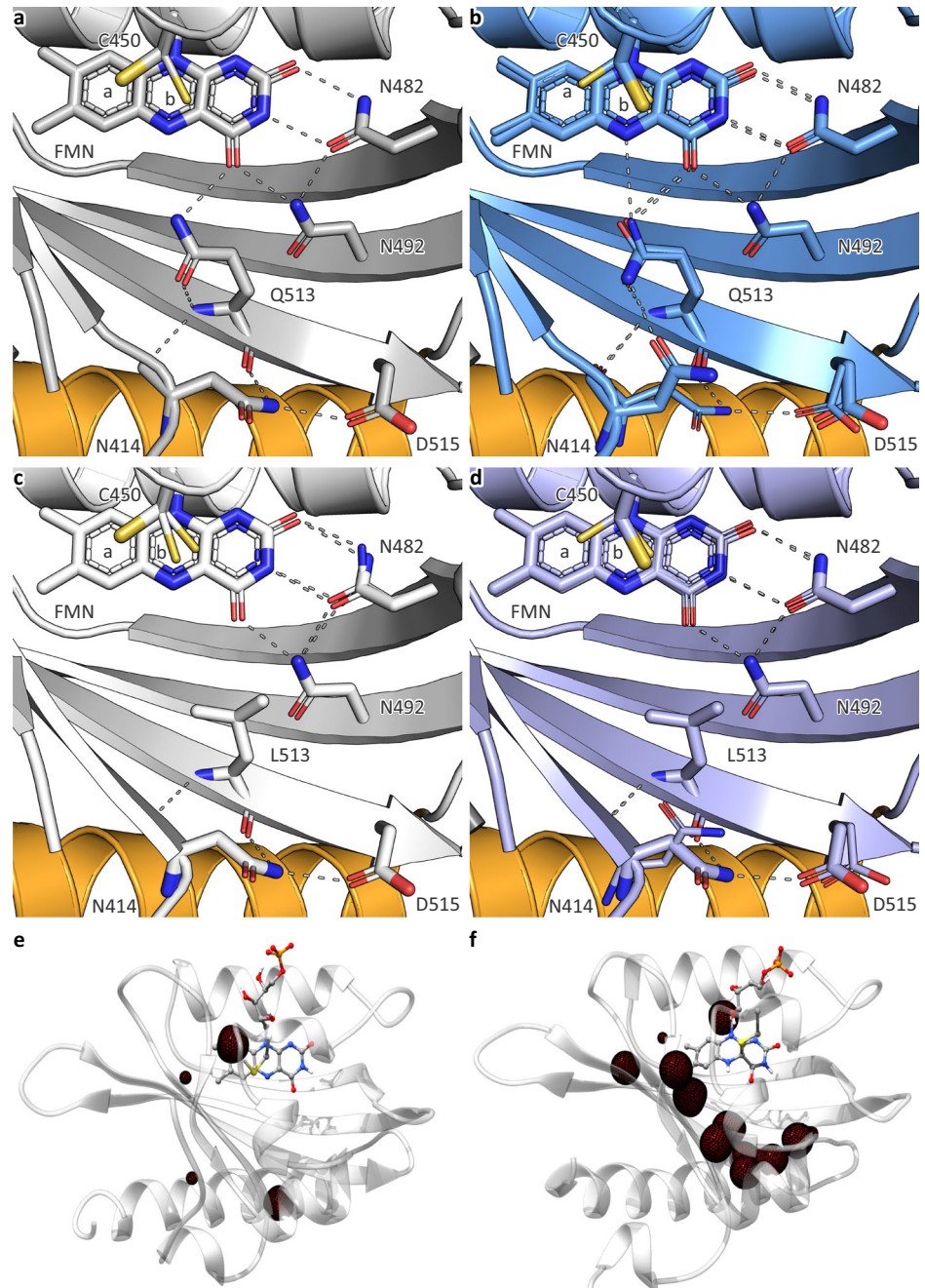

**Fig. 5 Structural analyses of *As*LOV2 variants. a** Chromophore-binding pocket of wild-type *As*LOV2 in its dark-adapted state as revealed by a 1.00 Å crystal structure. **b** Chromophore-binding pocket of wild-type *As*LOV2 in its light-adapted state as revealed by a 1.09 Å crystal structure. **c** Chromophore-binding pocket of *As*LOV2 Q513L in its dark-adapted state as revealed by a 0.90 Å crystal structure. **d** Chromophore-binding pocket of *As*LOV2 Q513L in its light-adapted state as revealed by a 0.98 Å crystal structure. For clarity, helices Cα and Dα are not shown in panels a-d. The Jα helix is drawn in orange, and the flavin- mononucleotide cofactor and key amino acids are highlighted in stick representation. Minor conformations of residues and the flavin nucleotide are drawn in narrower diameter. The active-site cysteine 450 adopts two principal orientations, denoted 'a' and 'b'. In the structures of dark-adapted *As*LOV2 wild-type, dark-adapted Q513L, and light-adapted Q513L, orientation 'b' splits into two subpopulations with slightly different $\chi_1$ angles. Dashed lines denote hydrogen bonds. **e** Water density in the interior of dark-adapted *As*LOV2 Q513L derived from a 300 ns classical molecular dynamics simulation. The red mesh denotes a density level of 0.3 water molecules per Å$^3$. **f** As **e** but for light-adapted *As*LOV2 Q513L. Corresponding simulations for *As*LOV2 wild-type are provided in Suppl. Fig. 13a, b.

chromophore-binding pocket and its surroundings (Fig. 5, Suppl. Fig. 7). Notably, these differences were consistent across several crystals, implying that they are genuinely tied to the Q513L exchange and illumination, respectively.

The structure of dark-adapted *As*LOV2 wild-type (Fig. 5a) well agreed with a previous determination at 1.4 Å (PDB entry 2v0u,

mainchain rmsd 0.13 Å)[47]. As observed before, the active-site cysteine 450 adopted a major (80%) conformation a, pointing away from the flavin C4a atom, and a minor (20%) one b, oriented towards C4a (Suppl. Figs. 7 and 8). The flavin pteridin moiety was coordinated by the asparagines N482 and N492, and the flavin O4 atom hydrogen-bonded to the amide NεH$_2$ group of

the conserved Q513. Via its $N\delta H_2$ group, N414 at the start of strand Aβ entered hydrogen bonds with the backbone carbonyl oxygen of Q513 and the carboxylate group of D515, situated at the tip of strand Iβ and part of the conserved PAS DIT motif[6]. At the present high resolution, an alternate conformation could be resolved for the terminal turn of the Jα helix (residues 543-546), possibly reflecting the inherent equilibrium between folded and unfolded helical states[15,56].

The light-adapted state of AsLOV2 wild type (Fig. 5b) exhibited a series of conformational differences consistent with a previous report at 1.7 Å resolution (PDB entry 2v0w, mainchain rmsd 0.21 Å)[47]. Given the higher resolution achieved presently, additional structural transitions could be pinpointed as summarized below. The sidechain of C450 reoriented towards the flavin C4a, thus shifting the ratio of the conformations a and b to 40%:60% (Suppl. Fig. 7). As in other structures of photoactivated LOV receptors[8,47], little electron density for the cysteinyl-flavin thioadduct was observed, likely owing to X-ray radiolysis of the metastable thioether bond. Beyond the altered conformation of C450, the population of the light-adapted state was indicated by a ~6.9° tilt of the isoalloxazine plane towards the cysteine (Suppl. Fig. 9)[8,19]. Based on earlier reports[8,19,47,55,57], chemical reasoning, and spectroscopic evidence[18,20], the sidechain of the conserved glutamine Q513 was modeled to undergo a 180° flip in response to enable hydrogen bonding between the amide Oε atom and the newly protonated flavin N5 position. Upon reorientation, the Q513 amide $N\epsilon H_2$ group hydrogen-bonded with the backbone carbonyl O of N414. The asparagine 414 in turn rotated, thus breaking contact to D515 and enabling a new hydrogen bond between its amide Oδ atom and $N\epsilon H_2$ of Q513 (Suppl. Fig. 7). Notably, the dark-state conformations of both Q513 and N414 were retained as a minor population (20%) in the light-adapted state, potentially due to incomplete photoactivation in the crystal. The reorientation of N414 correlated with a 0.4 Å shift of its Cα atom, thereby prompting the entire A'α segment to dislodge and move away from Q513 (Suppl. Fig. 10). Crucially, the A'α helix is interlocked with the C-terminal part of Jα via the hydrophobic residues L408, I411, I539, A542, and L546. The displacement of A'α thus went along with an outward movement of the last 1.5 helical turns of Jα, which could potentially promote its unfolding[15]. Support for this notion derives from the well-documented detrimental effect of the I539E substitution at the A'α:Jα interface[48], from molecular dynamics (MD) simulations that revealed correlated motions of the A'α and Jα helices[24], and from a recent study on circularly permuted AsLOV2 which pinpointed the Jα C-terminus as pivotal for light-dependent signaling, whereas the N-terminal part could be dispensed with[28]. In addition to the above differences, the light-adapted state also exhibited enhanced flexibility of the Aβ-Bβ and Gβ-Hβ loops, consistent with a global gain of mobility upon light absorption in AsLOV2 and other LOV domains[15,58].

In dark-adapted AsLOV2 Q513L (Fig. 5c), the flavin plane was displaced by ~0.4 Å relative to the wild-type protein, arguably due to steric interactions between the flavin O4 and the Cδ2 methyl group of L513. Notably, no ordered water molecules entered the space vacated by the glutamine removal. The resultant loss of hydrogen bonds at the flavin O4 atom may account for the hypsochromic absorbance shift evidenced above across the different LOV receptors with replaced glutamine. C450 adopted the orientations a and b, pointed away and towards the flavin C4a atom, respectively, at a ratio of 70%:30% (Suppl. Fig. 7). The Q513L replacement notwithstanding, the crucial N414 residue assumed the conformation seen in darkness for the wild type, i.e., engaged in hydrogen bonds with D515 and the backbone carbonyl O of residue 513. Interestingly, the Q513L dark state showed alternate conformations for the Aβ-Bβ and Gβ-Hβ loops,

in the case of the wild-type receptor only seen upon light exposure. Despite lacking the conserved glutamine, the AsLOV2 Q513L variant displayed structural responses in its light-state structure remarkably similar to the wild type, in line with the above functional assays that invariably demonstrated qualitatively intact light responses after leucine introduction. Specifically, C450 adopted the conformations a and b at a 40%:60% ratio, and the flavin ring plane tilted towards the cysteine by ~4.6°. Strikingly, L513 did not exhibit any dark-light differences, implying that its sidechain is inert and not actively participating in the signal relay. This notion is supported by the observation that most of the canonical amino acids with diverse sidechains supported productive light responses in the YF1 receptor (see Fig. 2b). Intriguingly, the crucial N414 assumed the light-adapted conformation to 40% extent; signals were evidently transduced from the flavin to this site even in the absence of the intermediary glutamine, if at reduced efficiency compared to wild-type AsLOV2. Rotation of the asparagine sidechain was accompanied by the same structural transitions evidenced in the wild-type receptor, most importantly an outward shift of the N414 Cα atom and the complete A'α segment (Suppl. Fig. 10).

Collectively, the data reveal at high resolution how light stimuli propagate from the flavin to the LOV β-sheet interface and the terminal A'α and Jα helices, structural elements generally associated with downstream signal transduction across LOV domains[15,32,46,59–61]. Strikingly, the Q513L variant underwent the same qualitative responses as the wild type which raises the question of how signal relay to N414 and beyond can be rationalized in the absence of the glutamine? As candidate mechanisms, we principally considered electrostatic interactions through space and water-mediated rearrangement of hydrogen-bonding networks. To assess the validity of these proposals, we resorted to molecular simulations. First, we evaluated how cysteinyl-flavin thioadduct formation and accompanying N5 protonation impact the electrostatic potential of wild-type AsLOV2 in the absence of other hydrogen-bonding changes. Electrostatics calculations revealed that changes in the potential were small in magnitude, largely confined to the immediate chromophore surroundings, and not extending far in space (Suppl. Fig. 11). Highly similar electrostatic potentials resulted for the corresponding AsLOV2 Q513L structures, and we thus deem signal transduction through space via altered electrostatics unlikely. Although the light-state structures of AsLOV2 wild-type and Q513L did not exhibit ordered water molecules in the immediate vicinity of position 513, we hypothesized that water might transiently enter the chromophore-binding pocket and thus relay the N5 protonation change in the light-adapted state. This notion finds support in classical MD simulations that indicate water penetration into the flavin binding pocket upon light exposure (Fig. 5e, f and Suppl. Fig. 12). Whereas in the simulations of dark-adapted AsLOV2 Q513L only two significant water clusters were observed inside the protein, the light-adapted state seemed to "soak" up water from the bulk solvent and displayed nine clusters in the protein interior. Closely similar results were obtained in simulations on AsLOV2 wild-type (Suppl. Fig. 13a, b). This striking phenomenon can be rationalized by reduced rigidity of the protein backbone upon formation of the cysteinyl adduct (Suppl. Fig. 13c, d). The pairwise root mean square deviation between snapshots from the MD trajectory was below 1.8 Å for dark-adapted AsLOV2 Q513L but lay in the region of 2.4 Å and higher for the light-adapted state. These findings concur with the above-mentioned increase in general protein mobility evidenced in LOV receptors upon thioadduct formation[15,58]. An overall increase in protein dynamics of the light-adapted state was also observed in earlier simulations[24]. As in the present study, the A'α:β-sheet interface

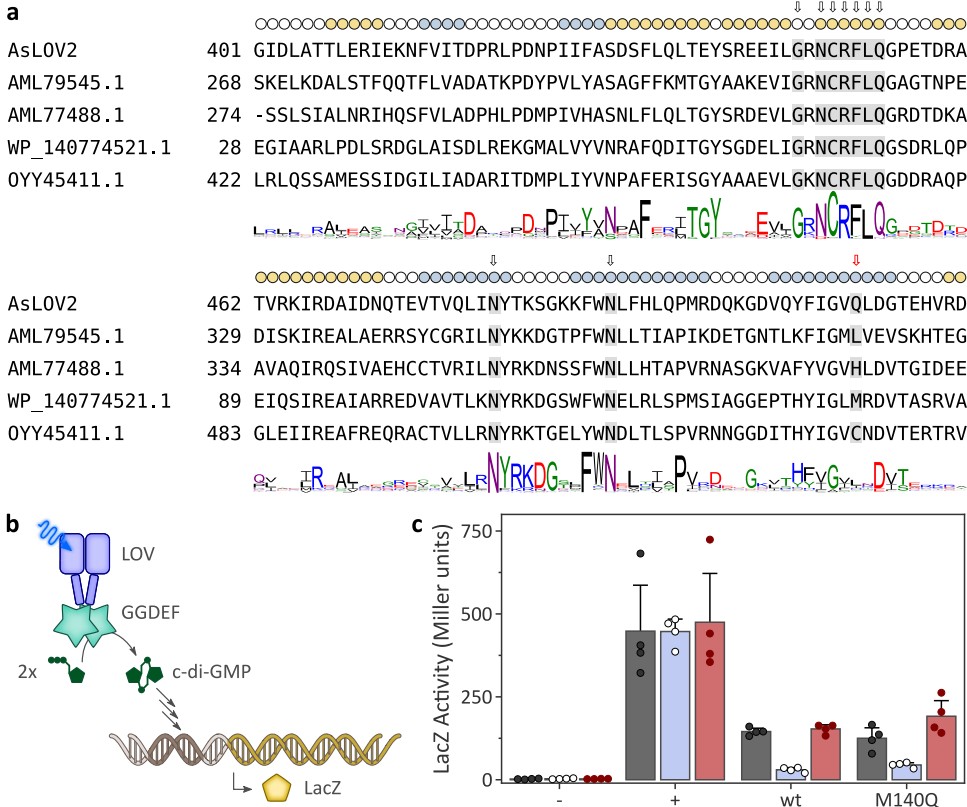

**Fig. 6 Naturally occurring, glutamine-deficient LOV$^{\Delta Q}$ receptors. a** Sequence searches identify around 350 receptors that have homology to bona fide LOV receptors but lack the conserved active-site glutamine. The multiple sequence alignment shows *As*LOV2 as a reference and four selected glutamine-deficient receptors. The sequence logo below the alignment was calculated for the entire set of glutamine-deficient LOV receptors (see Suppl. Data 3). Coloring, shading, and arrows as in Fig. 1, with the position of the conserved glutamine residue indicated by a red arrow. **b** Activity and light response of the LOV$^{\Delta Q}$-GGDEF fragment of WP_140774521.1 were assessed in an *E. coli* reporter strain harboring a *dgcE* knockout and a translational fusion between the cyclic-di-GMP-controlled *csgB* and *lacZ*. Light-dependent diguanylate cyclase activity can hence be gauged by measuring β-galactosidase levels. **c** Bacteria expressing the wild-type LOV$^{\Delta Q}$-GGDEF receptor or the M140Q variant were cultivated in darkness (black dots and gray bars), under blue light (white dots and blue bars), or under red light (red dots and bars). '−' refers to an empty-vector negative control, and '+' denotes a strain expressing the major diguanylate cyclase DgcE that served as the positive control. β-galactosidase activity is reported in Miller units and represents mean ± s.d. of four biologically independent replicates. The experiment was repeated twice with a similar outcome.

and residues N414 and Q513 were identified as crucial conduits for signal propagation. Consistent with our simulations, water was found to transiently enter the chromophore-binding pocket upon flavin thioadduct formation and N5 protonation. By contrast, the study reported reduced water influx and attenuated signal propagation in the case of the Q513L variant.

**Signal transduction in natural glutamine-deficient LOV receptors**. Given that LOV signal transduction evidently does not strictly depend on the conserved glutamine, we wondered whether LOV-like receptors exist in nature that lack this residue. To address this question, we conducted sequence searches and identified around 350 putative LOV receptors, denoted LOV$^{\Delta Q}$ in the following, that possess several residues highly conserved across LOV domains[62] but lack the active-site glutamine (Fig. 6a and Suppl. Data 3). Interestingly, these receptors featured a range of other amino acids in lieu of the active-site glutamine, predominantly the hydrophobic amino acids leucine and isoleucine, but also polar residues such as serine or threonine, and even histidine and cysteine. By contrast, large aromatic residues (phenylalanine, tyrosine, and tryptophan) were largely absent, as were proline and charged amino acids (Suppl. Fig. 14).

The sheer existence of LOV$^{\Delta Q}$ proteins in nature raises the tantalizing prospect that they can truly serve as blue-light

receptors. To principally address this possibility, we selected for further analysis a LOV$^{\Delta Q}$-GGDEF-EAL receptor from the proteobacterium *Mesorhizobium loti* which features a methionine at position 140 instead of the conserved glutamine (Genbank entry WP_140774521.1, see Fig. 6a). GGDEF and EAL domains antagonistically synthesize and degrade, respectively, the ubiquitous bacterial second messenger cyclic-di-(3'-5')-guanosine monophosphate (c-di-GMP)[63]. To assess potential light responses, we expressed the C-terminally truncated LOV$^{\Delta Q}$-GGDEF receptor in the *E. coli* reporter strain KN78 which lacks the major diguanylate cyclase DgcE and carries a translational fusion between the c-di-GMP-controlled *csgB* locus and β-galactosidase[64] (Fig. 6b). Bacteria were cultivated in darkness, under blue light, or under red light, and β-galactosidase activity was determined. As a positive control, a strain expressing DgcE exhibited constitutively high activity of around 450 Miller units (M.u.), irrespective of illumination (Fig. 6c). The KN78 strain carrying an empty plasmid served as a negative control and showed low activity of around 2 M.u., again independent of light. LOV$^{\Delta Q}$-GGDEF expression resulted in 145 M.u. in darkness but only 30 M.u. under blue light. Conversely, red light had no effect on the detectable activity. Replacement of M140 by glutamine yielded activity levels and light responses similar to those of the wild-type protein. Taken together, the results suggest that the *M. loti* LOV$^{\Delta Q}$-GGDEF protein acts as a blue-light-repressed

diguanylate cyclase despite lacking the conserved glutamine residue.

## Discussion

**Mechanism of signal transduction *sans* glutamine.** Following the description of LOV receptors as blue-light-receptive flavoproteins[1], optical and nuclear magnetic resonance spectroscopy identified the formation of the cysteinyl-flavin adduct in the signaling state[10,18]. Owing to a hybridization change of the flavin C4a atom from $sp^2$ to $sp^3$ in the adduct, the adjacent N5 atom is protonated and thus converted from a hydrogen-bond acceptor in the dark-adapted state to a donor in the signaling state (Suppl. Fig. 15). N5 protonation is an essential step in signal transduction as not least evidenced by the reconstitution of LOV receptors with 5-deaza-FMN[65]. Despite retaining the ability to form the thioadduct under blue light, these receptors are incapable of downstream signaling responses, arguably due to a lack of hydrogen bonding at the C5 position. Further support for the pivotal role of N5 protonation derives from cysteine-deficient LOV receptors that undergo photoreduction to the NSQ state which is protonated at N5 and thus elicits intact signaling responses[14]. Three-dimensional structures of phototropin LOV domains early on pinpointed the conserved glutamine residue close-by the flavin chromophore and in hydrogen-bonding distance to the O4 and N5 atoms[7,8,19]. Supported by spectroscopic evidence[18,20,66], the glutamine is generally held to rotate its sidechain upon N5 protonation to satisfy hydrogen bonding[8,19]. Possibly, this rotation is aided by transient rearrangements of two conserved asparagines (residues N482 and N492 in *As*LOV2, see Fig. 5) that coordinate the flavin- nucleotide chromophore[25,65]. Reorientation of the glutamine residue in turn provokes a cascade of hydrogen-bonding and structural changes, as for instance revealed in the past[47] and present structures of light-adapted *As*LOV2 (see Fig. 5). Photochemical reactions within the flavin chromophore, i.e., thioadduct formation or reduction to the NSQ state[14], are thus coupled to the protein scaffold, in particular, the LOV β-sheet and elements contacting it, e.g., N- and C-terminal extensions to the core domain. In *As*LOV2 specifically, asparagine 414 responds with a sidechain flip, accompanied by a shift of the protein backbone. Signals are thus channeled to the A'α and Jα helices and likely drive their light-dependent unfolding.

Irrespective of the strong conservation of the glutamine and its central involvement in canonical LOV signal transduction, its removal unexpectedly does not abolish light-dependent signaling responses. Intriguingly, this effect spans LOV receptors of distant phylogenetic origin and with disparate associated output modules (see Figs. 2–5), that invariably retained intact responses upon replacement of the glutamine, if to different and often reduced quantitative extent. In line with these observations, two recent reports revealed that the LOV domains from *Vaucheria frigida* aureochrome 1 and *A. thaliana* ZTL also elicited intact downstream responses after replacement of the glutamine by leucine or other residues[26,27]. Taken together, we propose that the conserved glutamine, long considered essential for LOV signal transduction, is in fact generally dispensable. This view is corroborated by the existence of hundreds of glutamine-deficient LOV$^{\Delta Q}$ proteins in nature (see Fig. 6a, Suppl. Data 3 and[26]), which presumably serve as blue-light receptors, as we presently demonstrate for a proteobacterial LOV$^{\Delta Q}$-GGDEF protein (see Fig. 6b).

Our functional and structural data suggest a potential mechanism for signal transduction in glutamine-deficient LOV receptors. The observation that most amino acids can stand in for the glutamine and support intact signal transduction (see Figs. 2 and 5) immediately argues against a direct involvement of the sidechain of these residues. Strikingly, the crystal structures of *As*LOV2 wild-type and Q513L revealed highly similar light-induced conformational changes that culminated in reorientation and altered hydrogen bonding of N414 and translocation of the A'α segment. The problem of signal transduction in glutamine-deficient LOV receptors thus reduces to the question of how signals are relayed across 10 Å from the newly protonated N5 atom to the LOV β-sheet, and specifically to N414 in *As*LOV2. In the following, we principally consider and discuss in turn as potential mechanisms i. steric rearrangements near the chromophore; ii. altered electrostatics in the thioadduct state; and iii. water-mediated hydrogen-bonding changes. First, as recently proposed for *A. thaliana* ZTL[26], steric rearrangements upon adduct formation, i.e., bond strain, $sp^2 \rightarrow sp^3$ hybridization change of the C4a atom, and tilting of the isoalloxazine heterocyclic system[8,19], might underpin signal propagation. However, the light-state Q513L structure did not reveal substantial conformational changes of residues immediately next to the flavin. Moreover, as previously demonstrated[14], cysteine-deficient LOV receptors can elicit canonical signaling responses when photoreduced to their NSQ state which is protonated at the N5 position like the thioadduct but experiences different steric constraints. Taken together, we thus regard steric effects as an unlikely general mechanism for signal propagation in glutamine-deficient receptors but note that for specific LOV proteins they plausibly play a crucial role[26]. Second, the formation of the thioadduct evidently modifies the electronic structure of the flavin and gives rise to an altered electrostatic potential. However, molecular simulations revealed (see Fig. 5) that such changes in electrostatics are comparatively small and of short reach. We hence deem it unlikely that electrostatic interactions transmitted through space are causative for signal transduction. Rather, we favor the third option of water-mediated hydrogen-bonding rearrangements, as illustrated in Fig. 7.

We propose that water molecules transiently enter the flavin-binding pocket, occupy the space vacated by glutamine removal, and form hydrogen bonds to the protonated flavin N5 and N414. Water would thus substitute for the glutamine side chain of canonical LOV receptors and relay hydrogen-bonding changes originating at the chromophore to the LOV β-sheet, and N414 in the case of *As*LOV2. We note that neither the dark-adapted nor the light-adapted structures of *As*LOV2 Q513L revealed direct evidence for ordered water molecules near the flavin N5 atom. However, support for our model derives from MD simulations suggesting that water dynamically enters this region of the light-adapted receptor. Moreover, the model would explain why, as one of only a few amino acids, proline cannot functionally substitute for glutamine, despite leaving chromophore binding and LOV photochemistry intact. In the imino acid proline, the Cγ and Cδ methylene groups of the sidechain loop back onto the amide nitrogen atom, thus sterically interfering with the proposed water-mediated hydrogen bonding. Alternatively, we cannot however rule out that proline fails to convey light signals because of its restricted conformational freedom or its lack of an amide proton. Lastly, the proposed mechanism would rationalize the near-identical conformational changes elicited by light in both *As*LOV2 wild-type and Q513L. Regardless of the presence of the glutamine, light signals would initially be converted into altered flavin N5 protonation and a subsequent hydrogen-bonding cascade that propagates to N414 at the LOV β-sheet interface[47]. Concomitant with the formation of new hydrogen bonds, N414 would break or weaken the hydrogen bonds formed in darkness between its backbone oxygen and the amide proton of residue 513, and between its NδH₂ amide group and the sidechain of D515, respectively. The resultant weakening of the LOV β-sheet would then transmit to the A'α and Jα helices that interact with the outer face of the sheet.

**Fig. 7 Signal transduction in light-oxygen-voltage (LOV) receptors lacking the conserved glutamine, exemplified for the *A. sativa* phototropin 1 LOV2 domain. a** Lewis formulae show the flavin- nucleotide chromophore and surrounding residues of the glutamine-deficient leucine variant in the dark-adapted (left) and light-adapted states (right). As revealed by X-ray crystallography (see Fig. 5), qualitatively similar structural responses to light-induced N5 protonation (see Fig. 1) are observed in both the absence of the conserved glutamine Q513 and in its presence (see Suppl. Fig. 15). Without the glutamine, water molecules might transiently enter the chromophore-binding pocket, thereby stand-in for the glutamine, and relay the signal as changes in hydrogen bonding to the Iβ and Aβ strands of the central β pleated sheet (involving residues N414 and 513). Notably, signals are thus also propagated to the LOV C terminus (D515) which is frequently engaged in signal transduction, and often exhibits a conserved DIT motif[6]. **b** The observation that LOV receptors can transduce signals without either or both of their strongly conserved cysteine and glutamine residues suggests a potential origin from redox-active flavoproteins[14]. LOV signal transduction in a primordial LOV ancestor lacking the Cys and Gln residues would have relied on flavin photoreduction to the NSQ radical and on water mediation. Both the Cys and Gln residues would be secondary acquisitions that minimize side reactions (Cys); enhance the fidelity of signal transduction (Cys and Gln); bathochromically shift the action spectrum (Gln); accelerate the dark recovery and thereby benefit temporal resolution (Cys and Gln); and render the signaling state less susceptible to the cellular environment (Cys). Note that we have no evidence in which sequential order the Gln and Cys residues may have been acquired.

Although residue N414 is not strictly conserved (see, e.g., Fig. 1 and Suppl. Fig. 1), the proposed model of signal transmission principally extends to other LOV receptors. Even in the absence of a polar residue at the position equivalent to N414, hydrogen-bond rearrangements could still be relayed to the β-sheet and beyond, as for instance evidenced in *Neurospora crassa* Vivid[22,57]. Across several LOV receptors, the outer β-sheet face and the adjacent DIT motif[6] recurrently take center stage in signal transduction[15,33,46,59,61,67]. Once relayed there, signals are then channeled into disparate structural responses in individual LOV receptors, including order-disorder transitions, association reactions, and quaternary structural transitions[5]. It is worth noting that our experimental data, simulations, and mechanistic proposal are not in contradiction with common models advanced for signal transition in the presence of glutamine. For instance, a recent study suggests that two conserved asparagine residues aid light-triggered glutamine reorientation[25], which would be compatible with our model (see Fig. 5 and Suppl. Fig. 15). By principally rationalizing how signal transduction occurs in the absence of glutamine, our model reinforces the central roles of N5 protonation and hydrogen bonding in LOV signal transduction. These core aspects likely apply to receptors with intact glutamine as well, as also suggested by earlier molecular simulations[23,24].

**LOV passes the QC.** Our data demonstrate that LOV receptors can evidently transduce light signals without the conserved glutamine. As qualitatively intact light responses are evoked upon glutamine replacement across all systems tested, we consider signaling in the absence of the glutamine a general and inherent, yet dormant trait of LOV receptors. This view is borne out by the existence of numerous glutamine-deficient LOV$^{\Delta Q}$ receptors in nature that could potentially serve as bona fide blue-light receptors. In a similar vein, we previously showed that LOV$^{\Delta C}$ receptors devoid of the conserved cysteine exist in nature and can elicit productive light responses owing to photoreduction to the NSQ state which is protonated at the flavin N5 atom[14]. We show presently that the paradigm *As*LOV2 domain perplexingly retains signaling capability, if at greatly attenuated efficiency, even when both the conserved cysteine and glutamine are replaced. Building on our earlier proposal[14], these observations jointly raise the prospect that LOV receptors arose during evolution from originally light-inert flavoproteins, e.g., enzymes involved in redox processes (Fig. 7b). The question then begs, if signal transduction can take place in the absence of cysteine and glutamine, why are these residues so prevalent in recent LOV receptors? Our data provide clues as to the potential driving forces underlying the strong glutamine conservation. First, the introduction of

glutamine generally enhances the fidelity and degree of the light response. Second, glutamine induces a bathochromic absorbance shift of ~10 nm, thus expanding light sensitivity to longer wavelengths. Third, glutamine accelerates the base-catalyzed dark recovery reaction[66], thus enhancing temporal resolution of light-dependent physiological responses. Similarly, the cysteine may have prevailed as its introduction minimizes side reactions (fluorescence and photosensitizing), desensitizes the light-adapted signaling state against environmental influences (e.g., partial oxygen pressure and redox conditions), and enhances the fidelity of the signaling response[14].

Beyond implications for the potential origin of LOV receptors, our data directly pertain to applications in optogenetics and biotechnology. First, replacement of the conserved glutamine residue generally decelerated the dark- recovery kinetics but preserved signaling responses to a substantial extent. Targeted modification of the glutamine residue thus provides a so-far little-explored avenue towards modulating these kinetics and thus the effective light sensitivity at photostationary state (see Suppl. Fig. 3)[39,40]. In a similar vein, glutamine substitution may serve to deliberately attenuate the light response as demanded by the application. Second, substitutions of either the conserved cysteine or glutamine residues have often been used as presumably light-insensitive, unresponsive negative controls. Our data however illustrate that even when these residues are replaced, LOV receptors can principally transduce light signals, although likely with reduced amplitude. These considerations transcend the optogenetic deployment of LOV receptors and also concern the widespread applications of cysteine-deficient (and often additionally glutamine-deficient[68]) LOV modules as fluorescent proteins[69,70] and photosensitizers for molecular oxygen[53].

## Methods

**Molecular biology**. YF1 variants with residue Q123 replaced were constructed in the background of the pDusk-DsRed and pDawn-DsRed reporter plasmids[31], or the expression plasmid pET-41a-YF1[32] by PCR amplification and blunt-end ligation. The gene of the cognate response regulator BjFixJ from Bradyrhizobium diazoefficiens, formerly designated B. japonicum, was amplified from an earlier expression construct[33], subcloned onto the pET-19b vector (Novagen), and thus furnished with an N-terminal His₆-SUMO tag. Substitutions of residue Q347 in the NmPAL receptor were performed according to the QuikChange protocol (Agilent Technologies) in either the pCDF-PALopt reporter plasmid or the pET-28c-PALopt expression plasmid[46]. For the expression of AsLOV2, a gene encoding residues 404–546 of A. sativa phototropin 1 (Uniprot O49003) was synthesized with an N-terminal GEF extension[15,47] and codon usage adapted to E. coli (GeneArt, ThermoFisher), and was cloned into the pET-19b vector. AsLOV2 was thus equipped with an N-terminal His₆-SUMO tag and its expression was put under the control of a T7-lacO promoter. Replacements of the active-site residues Q513 and C450 were generated by Quik-Change. Deletions of the N- and C-terminal A'α and Jα helices were prepared by PCR amplification and blunt-end ligation of the vector; the resultant truncated AsLOV2 variant comprised residues 411–517. The gene encoding residues 1-326 of the glutamine-deficient LOV-GGDEF receptor (ANN58260.1/WP_140774521.1) was amplified by PCR from genomic DNA of the proteobacterium Mesorhizobium loti NZP2037 (purchased from Deutsche Sammlung für Mikroorganismen und Zellkulturen, DSMZ no. 2627) and confirmed by DNA sequencing. A version of the gene with codons optimized for E. coli (GeneArt) was cloned into the pQE-30 vector (Qiagen) via Gibson cloning[71]. Residue replacements were prepared by PCR amplification and blunt-end ligation. All oligonucleotide primers (Suppl. Table 3) were purchased from Integrated DNA Technologies. All constructs were verified by Sanger sequencing (Microsynth AG, Göttingen).

**Protein expression and purification**. Protein expression and purification were carried out as previously described for YF1[32] and NmPAL[46]. To express and purify the response regulator BjFixJ, the above pET-19b BjFixJ expression plasmid was transformed into E. coli BL21 CmpX13 cells[72]. Bacteria were grown at 37 °C in Luria broth (LB) medium to an optical density at 600 nm ($OD_{600}$) of around 0.6-0.8, at which point the temperature was lowered to 16 °C and expression induced by the addition of 1 mM isopropyl β-D-1-thiogalactopyranoside (IPTG). Following incubation overnight at 16 °C, cells were lysed by sonication, and the supernatant was cleared by centrifugation and purified by Ni²⁺ immobilized metal ion affinity chromatography (IMAC). The His₆-SUMO tag was cleaved off by the SUMO protease Senp2, followed by a second IMAC purification. BjFixJ protein was dialyzed into storage buffer [20 mM tris(hydroxymethyl)aminomethane (Tris)/HCl pH 8.0, 250 mM NaCl,

10% (w/v) glycerol], and the concentration was determined using an extinction coefficient of 4860 M⁻¹ cm⁻¹ at 280 nm[33].

For the production of AsLOV2 variants, the pET-19b expression plasmid (see above) was transformed into E. coli BL21 CmpX13 or LOBSTR cells[73]. Protein expression was induced by the addition of 1 mM IPTG and conducted at 16 °C overnight. When using the CmpX13 strain, the medium was supplemented with 50 μM riboflavin. The cleared bacterial cell lysate was purified by Co²⁺ IMAC, Senp2 cleavage of the His₆-SUMO tag, and a second IMAC step, as described for BjFixJ. Depending on purity, AsLOV2 variants were further purified by anion-exchange chromatography. Purified protein was dialyzed into storage buffer [20 mM Tris/HCl pH 7.4, 20 mM NaCl, 20% (v/v) glycerol], and its concentration was determined spectroscopically using an extinction coefficient of 13,800 M⁻¹ cm⁻¹ for the flavin absorption maximum around 447 nm[10].

**Spectroscopic analyses**. UV/vis absorbance spectra were recorded on an Agilent 8435 diode-array spectrophotometer at 22 °C, as controlled by an Agilent 89090 A Peltier thermostat. Absorbance spectra were acquired for the dark-adapted LOV receptors and after saturating illumination with a 455-nm light-emitting diode (LED) (30 mW cm⁻²) for the light-adapted states. Throughout all light intensities were determined with a power meter (model 842-PE, Newport) and a silicon photodetector (model 918D-UV-OD3, Newport). The recovery to the dark-adapted state was monitored by recording spectra over time. The resultant kinetics were corrected for baseline drift and evaluated by nonlinear least-squares fitting to exponential functions using the Fit-o-mat software[74]. Absorbance spectroscopy on YF1 variants was conducted at 37 °C in 20 mM Tris/HCl pH 8.0, 20 mM NaCl; to accelerate the recovery in the Q123L variant, up to 1 M imidazole was added[13], and the resulting rate constants for dark recovery were extrapolated to 0 M imidazole. UV/vis-spectroscopic analysis of NmPAL was performed in 12 mM 4-(2-hydroxyethyl)−1-piperazineethanesulfonic acid (HEPES)/HCl pH 7.7, 135 mM KCl, 10 mM NaCl, 1 mM MgCl₂, 10% (v/v) glycerol[46]. AsLOV2 variants were analyzed in 10 mM sodium phosphate pH 7.5, 10 mM NaCl; to aid solubility, for the Q513D variant 20% (v/v) glycerol was added. To promote photoreduction in the cysteine-devoid AsLOV2 C450A variant, 1 mM tris(2-carboxyethyl)phosphine (TCEP) was added.

Secondary structure and light-induced changes were assessed by CD spectroscopy on a JASCO J710 spectrophotometer equipped with a PTC-348WI Peltier element. CD spectra were recorded at 22 °C in a 1-mm cuvette for the dark-adapted state and following saturating blue-light illumination for the light-adapted state. All spectra were corrected by blank spectra and represent the average of at least 4 scans. In the case of the faster-recovering AsLOV2 variants, blue light was applied before each scan. Buffers were as above except for NmPAL where 12 mM sodium phosphate pH 7.7, 135 mM KCl, 10 mM NaCl, 1 mM MgCl₂, 10% (v/v) glyerol was used instead. In the case of the AsLOV2 variants, the return to the dark-adapted state after blue-light exposure was monitored over time at a wavelength of (208 ± 5) nm and evaluated by fitting to exponential functions using Fit-o-mat[74].

**YF1 functional assays**. The net kinase activity of YF1 variants and its dependence on blue light were assessed in the pDusk-DsRed reporter setup[31,75]. To this end, pDusk-DsRed plasmids harboring different YF1 variants were transformed into E. coli CmpX13. Individual wells of a 96-deep-well microtiter plate (P-DW-11-C-S, Corning, New York) containing 400 μL LB supplemented with 50 μg mL⁻¹ kanamycin were inoculated with a given YF1 variant. Plates were sealed with a gas-permeable membrane (BF-410400-S, Corning) and incubated for 16 h at 37 °C and 700 rpm in either darkness or under constant blue light (470 nm, 100 μW cm⁻²). Following incubation, $OD_{600}$ and the fluorescence of the DsRed Express2 reporter[76] were measured with a Tecan Infinite M200 PRO plate reader (Tecan Group Ltd. Männedorf, Switzerland). For the fluorescence measurements, the excitation wavelength was (554 ± 9) nm and that of the emission (591 ± 20) nm. Fluorescence data were divided by $OD_{600}$ and normalized to the value for YF1 under dark conditions. Data represent the mean ± s.d. of three biologically independent samples. The response to trains of blue-light pulses was assessed for pDawn-DsRed systems harboring different YF1 variants as previously described[41]. Briefly, bacterial cultures were grown in sealed, black-walled 96-well microtiter plates (Greiner BioOne, Frickenhausen, Germany) for 16 h at 37 °C and 600 rpm. The transparent bottom of the plates allowed illumination from below with a programmable matrix of LEDs. Following incubation, $OD_{600}$ and DsRed fluorescence were measured and evaluated as above.

Activity and light response of purified YF1 variants were characterized in a coupled assay that reports on the phosphorylation-induced binding of BjFixJ to a fluorescently labeled, double-stranded DNA (dsDNA). To this end, a dsDNA substrate with the sequence 5'-GAG CGA TAT CTT AAG GGG GGT GCC TTA CGT AGA ACC C-3' and labeled at its 5' end with (5-and-6)-carboxytetramethylrhodamine (TAMRA) was prepared as described before[14]. The underlined portion of the sequence corresponds to the BjFixK2 operator site that BjFixJ binds to[43]. To assess light-dependent catalytic activity, 2.5 μM of each YF1 variant in its dark-adapted state were incubated at 25 °C with 1.25 μM BjFixK2 dsDNA substrate and 25 μM BjFixJ in buffer containing 10 mM HEPES/HCl pH 7.6, 80 mM KCl, 2.5 mM MgCl₂, 0.1 mM ethylenediaminetetraacetic acid, 111 μg mL⁻¹ bovine serum albumin (BSA), 10% (v/v) glycerol, 4% (v/v) ethylene glycol and 20 mM TCEP. The solution was transferred to a black 96-well microtiter

plate (FluoroNunc). Upon starting the reaction with the addition of 1 mM ATP, the kinetics were followed by measuring TAMRA fluorescence anisotropy with a multi-mode microplate reader (CLARIOstar, BMG Labtech) over 30 min. Fluorescence was recorded at excitation and emission wavelengths of (540 ± 10) nm and (590 ± 10) nm, respectively, and using a 566-nm long-pass beam splitter. After 30 min, the microtiter plate was ejected, the samples were illuminated for 30 s with a 470-nm LED (30 mW cm$^{-2}$), and the measurement continued for another 12 min.

**NmPAL functional assays**. The light-dependent binding of NmPAL variants to their RNA target was assessed in a bacterial reporter-gene system[46]. Briefly, E. coli CmpX13 cells[72] were transformed with the arabinose-inducible pCDF-PALopt expression and the pET-28c-DsRed-SP reporter plasmids[46]. Notably, the reporter plasmid contains the NmPAL aptamer 04.17 upstream of the Shine-Dalgarno (SD) sequence of the DsRed gene; NmPAL binding to this site thus reduces reporter expression at the mRNA level. Bacterial starter cultures were grown at 37 °C overnight, transferred to individual wells of a 96-deep-well microtiter plate, and diluted to an $OD_{600}$ of 0.03 in 700 µL LB medium supplemented with 4 mM arabinose, 50 µg mL$^{-1}$ kanamycin, and 100 mg mL$^{-1}$ streptomycin. Following 2 h incubation at 37 °C and 600 rpm, cultures were supplemented with 1 mM IPTG to induce DsRed expression. Cultures were then split into two samples which were incubated for 16 h at 29 °C in darkness or under blue light (470 nm, 40 µW cm$^{-2}$), respectively. $OD_{600}$ and DsRed fluorescence were determined as described above. Data represent the mean ± s.d. of four biologically independent replicates.

For the quantitative analysis of NmPAL binding to RNA, we recorded its interaction with 4 nM TAMRA-labeled 04.17 aptamer by fluorescence anisotropy as described before[46]. Experiments were carried out in reaction buffer containing 12 mM HEPES/HCl pH 7.7, 135 mM KCl, 10 mM NaCl, 1 mM MgCl$_2$, 10% (v/v) glycerol, 100 µg mL$^{-1}$ BSA. Fluorescence anisotropy was recorded with a multi-mode microplate reader (CLARIOstar) at (540 ± 10) nm excitation, (590 ± 10) nm emission, and using a 566-nm long-pass beam splitter. Data obtained in the presence of rising concentrations of either dark-adapted or light-adapted NmPAL (obtained by illumination with 455 nm, 50 mW cm$^{-2}$, 60 s) were fitted to single-site binding isotherms using Fit-o-mat[74] according to Eq. (1).

$$r = r_0 + r_1 \times [PAL]/([PAL] + K_d) \tag{1}$$

To probe the light-dependent activity of NmPAL variants in eukaryotic cells, 50,000 Hela cells per well were seeded in 24-well plate format[46]. Following 24 h incubation at 37 °C, cells were transfected. In brief, the medium was aspirated, and 500 µL OptiMem medium were added to each well. In parallel, the transfection mix was prepared by combining 450 ng plasmid encoding an mCherry-tagged NmPAL variant and 50 ng reporter plasmid encoding Metridia secreted luciferase in 50 µL OptiMem plus 2 µL lipofectamin 2000. Upon incubation for 20 min at room temperature, 50 µL of the transfection mix were added to each well, followed by incubation for 4 h at 37 °C in either darkness or under blue light (100 µW cm$^{-2}$, 465 nm, 60 s dark intervals followed by 30 s light intervals). The cell supernatant was then replaced by full medium (DMEM, supplemented with 10% fetal calf serum), and incubation continued at 37 °C. At 19 h post transfection, the luciferase expression was assessed by transferring 50 µL of the cell supernatant to a fresh 96-well white plate (Lumitrac 200, Greiner). 5 µL of the luciferase reagent (Ready-To-Glow secreted luciferase, Takara Clontech) were added to each well, and the plate was incubated for 25 min at room temperature. Chemiluminescence was then measured using an EnSpire plate reader (Perkin Elmer) with an integration time of 5 s.

**Diguanylate cyclase assay**. The activity of the LOV-GGDEF protein was assessed in the E. coli strain KN78 which carries a knockout of the major diguanylate cyclase DgcE and encodes in its genome a translational fusion between the nucleator protein csgB involved in curli formation and the β-galactosidase lacZ[64,77]. To this end, a pQE-30 vector encoding a given LOV-GGDEF variant was transformed into E. coli. An empty pQE-30 plasmid served as negative control; as a positive control, the empty pQE-30 plasmid was transformed into strain AR1100 which expresses a functional copy of DgcE. Bacterial starter cultures were grown overnight at 37 °C in 5 mL LB medium supplemented with 50 µg mL$^{-1}$ ampicillin. Cultures were then diluted 100-fold, 1 mM IPTG was added, and growth continued for 24 h at 28 °C and 550 rpm in either darkness, under constant blue light (450 nm, 40 µW cm$^{-2}$), or under constant red light (660 nm, 40 µW cm$^{-2}$). LacZ activity was then determined according to Miller[78] using the chromogenic substrate ortho-nitrophenyl-β-galactoside. Data represent mean ± s.d. of three separate experiments comprising four biologically independent replicates each.

**Structure determination of AsLOV2 variants**. The expression vectors for the AsLOV2 variants were intentionally designed such that upon Senp2 cleavage during purification (see above) the same N-terminal GEF cloning artifact resulted as in a previous structural study[47]. Crystallization was conducted by sitting-drop vapor diffusion at solvent conditions adapted from the previous report[47]. Orthorhombic crystals were obtained at protein concentrations between 10 and 20 mg mL$^{-1}$ in 0.1 M sodium acetate pH 4.6–5.0, 6–8% (w/v) PEG 4000, 30% (v/v) glycerol. Crystal growth and handling were generally performed in darkness or under dim red light, respectively. To characterize the dark-adapted state, single

crystals were mounted in loops and rapidly cryo-cooled by immersion in liquid nitrogen. To assess the light-adapted state, crystals were exposed to blue light (470 nm, 20 mW cm$^{-2}$, 1 min) prior to cryo-cooling. Diffraction data were collected at the BESSY synchrotron (beamlines 14.1 and 14.2)[79] to resolutions between 0.90 Å and 1.09 Å (Suppl. Tables 1 and 2).

Indexing and integration were performed with XDS[80], and scaling was done with Pointless[81], all through the XDSapp interface[82]. Structures were solved by molecular replacement using phaser[83], and with the previously determined structure of dark-adapted AsLOV2 as the search model (PDB entry 2v0u[47]). Model building was done in Coot[84], and restrained refinement with anisotropic B factors was conducted in Refmac[85]. Occupancies of residues with multiple conformations were manually refined. Due to the absence of electron density for the covalent thioadduct in the light-adapted structures, the cofactors were generally modeled as noncovalently bound oxidized flavin mononucleotides. Atom coordinates and structure-factor amplitudes were deposited in the Protein Data Bank under accession codes 7pgx (wild-type, dark), 7pgy (wild-type, light), 7pgz (Q513L, dark), and 7ph0 (Q513L, light). Molecular graphics were prepared with PyMOL (Schrodinger LLC). The root mean square deviation between the structures was calculated with LSQKAB[86]. $2F_o−F_c$ omit maps (see Suppl. Fig. 7) were produced by setting the occupancies of atoms of interest to zero, followed by restrained refinement as described above.

To calculate light-dark difference electron density maps (see Suppl. Fig. 8), the datasets for the dark-adapted and light-adapted states of AsLOV2 wild-type and Q513L, respectively, were jointly scaled using Scaleit[87]. $F_{light}−F_{dark}$ difference maps were calculated with phase information of the respective dark-adapted state. Corresponding MTZ files are provided as Supplementary Data. In case of AsLOV2 Q513L, the best-resolved dark and light datasets (see Suppl. Table 2), differed in the b dimension of the unit cell by 1.5 Å. We, therefore, selected a different light dataset at a resolution of 1.24 Å in which this difference only amounted to 0.85 Å.

**Electrostatics and molecular simulations**. The crystal structures of AsLOV2 wild-type and Q513L determined in this work (PDB 7pgx and 7pgz) served as starting points for the simulation of the dark-adapted states. These structures were then modified to mimic the light-adapted state by forming a covalent bond between the Sγ atom of C450 and the C4a atom of the flavin residue. Further, a proton was transferred from the thiol group to the N5 atom of flavin. To specifically probe the effect of adduct formation and N5 protonation on electrostatics, the conformation of other residues was left unchanged. Subsequently, the protein structures were relaxed using a QM/MM geometry optimization. The electrostatic potential acting on a selected residue (e.g., N414) was calculated using the APBS 3.0 software[88]. The dielectric constants for inside/outside of the molecular regions were chosen to be 1.0 and 78.54, respectively. The solvent radius was set to 1.4 Å, and the temperature had the standard value of 298.15 K. The partial charges for all atoms were taken from the ff14SB force field of AMBER[89]. For each structure, four calculations were carried out to determine the electrostatic potential exerted by the protein environment on residues N414, N482, N492, and Q513/L513, respectively. The electrostatic potential maps were plotted with UCSF chimera[90] and are shown in Suppl. Fig. 11.

MD simulations were performed using the crystal structures obtained in this work. Missing hydrogen atoms were added to the initial structures using the tleap program of AMBER 18[89]. The protonation states of all titratable residues were considered at a pH of 7.0. The protein was solvated in a truncated octahedral box of TIP3P water molecules with a distance of at least 15 Å between the protein atoms and the boundaries of the box. The system was neutralized by adding K$^+$ and Cl$^−$ ions. The SHAKE algorithm[91] was used to constrain the bonds involving hydrogen atoms, thus allowing the time step to be 2 fs. A Langevin thermostat with a collision frequency of 1 ps$^{-1}$ was used for temperature control in all simulations. The VMD[92] plugin VolMap served to analyze the water density inside the protein. The MM parameters for FMN and the FMN-Cys adduct were obtained from[93]. Initially, the solvent was minimized in 100,000 steps with restraints of 100 kcal mol$^{-1}$ Å$^{-2}$ on all protein atoms and FMN. The system was gradually heated from 100 K to 300 K within 50 ns with restraints on the protein and FMN in NVT ensemble. The density of the solvent was then gradually equilibrated for another 20 ns under NPT conditions. The equilibration was extended for another 20 ns with weaker restraints of 10 kcal mol$^{-1}$ Å$^{-2}$. Then, MD of 20 ns each was conducted with weakened restraints of 1 kcal mol$^{-1}$ Å$^{-2}$ and 0.1 kcal mol$^{-1}$ Å$^{-2}$, respectively, on the protein backbone. Finally, an unrestrained MD production run of 300 ns was carried out.

**Sequence analysis of LOV receptors lacking the active-site glutamine**. As in a previous analysis[14], a BLAST search was performed with Bacillus subtilis YtvA (BsYtvA[94], residues 1-127) as the query sequence and with an E value cutoff of 10. Using a custom Python script (available from https://github.com/TheAngulion/LOVfilterQ), the results were filtered for entries that possess at least eight out of nine residues (residue positions Gly59, Asn61, Cys62, Arg63, Phe64, Leu65, Gln66, Asn94 and Asn104 in BsYtvA), which are highly conserved across LOV receptors[62], but lack the active-site glutamine (position Gln123 in BsYtvA). Corresponding entries were aligned to the sequences of BsYtvA and AsLOV2 using ClustalX[95]. A sequence logo was generated with WebLogo version 3.7[96].

**Reporting summary**. Further information on research design is available in the Nature Research Reporting Summary linked to this article.

## Data availability

Atom coordinates and structure-factor amplitudes have been deposited in the Protein Data Bank under accession codes 7pgx (*As*LOV2 wild-type, dark), 7pgy (wild-type, light), 7pgz (Q513L, dark), and 7ph0 (Q513L, light). MTZ files underpinning the light-dark difference electron density maps (see Suppl. Fig. 8) and a multiple sequence alignment of LOV$^{\Delta Q}$ receptors are provided as Supplementary Data. Other data generated in this study have been deposited in the Zenodo database under accession code 6341885.

## Code availability

A Python script for sequence analysis of glutamine-deficient LOV receptors is available from https://github.com/TheAngulion/LOVfilterQ.

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

## Acknowledgements

We thank U. Jenal and B. Zoltowski for discussion; R. Hengge for discussion and supplying the *E. coli* reporter strain harboring the genomic csgB::lacZ integration; and C. Xu for assistance with ChemDraw. Diffraction data were collected on BL14.1 and BL14.2 at the BESSY II electron storage ring operated by the Helmholtz-Zentrum Berlin[79], with the assistance of C. Feiler and F. Lennartz.

## Author contributions

J.D., R.G., and J.K. contributed equally. J.D. performed all experiments on the YF1 and LOV^ΔQ-GGDEF variants. R.G. performed all experiments on the isolated *As*LOV2 domain and refined crystal structures. J.K. analyzed *Nm*PAL in bacterial reporter assays and by absorbance spectroscopy, and she studied its RNA interaction by fluorescence anisotropy. V.B. and I.S. conducted and evaluated molecular simulations. C.R., S.P., and G.M. did experiments on *Nm*PAL in eukaryotic cells. A.T.R. performed spectroscopy on *Nm*PAL and analyzed RNA binding. A.G.F. analyzed *As*LOV2 variants by CD spectroscopy. T.G. and R.P.D. developed the fluorescence anisotropy assay for YF1. M.W. advised on crystallization and structure refinement. A.M. conducted sequence analyses, refined crystal structures, and conceived and coordinated the research. J.D. and A.M. wrote the manuscript with input from all authors.

## Funding

Funding by the Deutsche Forschungsgemeinschaft (DFG) (491183248; grants MO2192/6-1/2 and MO2192/8-1 to A.M.; grants MA3442/5-1/2 to G.M.) and the Alexander-von-Humboldt Foundation (Sofja-Kovalevskaya Award to A.M.) is gratefully acknowledged. I.S. acknowledges support by the DFG through SFB 1078, project C6. Open Access funding enabled and organized by Projekt DEAL, and supported by the DFG and the Open Access Publishing Fund of the University of Bayreuth.

## Competing interests

The authors declare no competing interests.
