## [Peer Review File · Nature Communications]

REVIEWER COMMENTS

Reviewer #1 (Remarks to the Author):

In their manuscript, Dietler and colleagues study signal transduction in LOV photoreceptors. These proteins are widespread in nature and are also used as tools in molecular biology. Their mechanism of action is thought to be well understood and more or less conserved among different LOV receptor classes.

However, the authors show that a conserved glutamine believed to mediate signal transduction from the flavin chromophore is not strictly required for signaling. The authors use functional assays to show that three different LOV proteins where the glutamine is mutated are still capable of signal transduction. Then, they identify 350 putative LOV proteins naturally lacking the glutamine in genomic databases and show that one of them is a functional photoreceptor.

The authors also use crystallography and molecular dynamics simulations to study signal transduction in natural and mutated AsLOV2 to gain insight into the possible mechanism of signal transduction in LOV domains lacking the glutamine, and then propose a theory of how the photoreceptors might have evolved from light-insensitive flavoproteins. The crystal structures are of a very high resolution, probably exceeding the best resolution reported in the literature for any LOV protein.

Generally, the manuscript is clearly and well written, and contains all the necessary details. While earlier studies hinted at some of the authors' findings, the authors here provide a complete and very detailed investigation. Given that the studied proteins belong to different classes and are from very different organisms, the finding is likely to be general and not a peculiarity of a subfamily of LOVs. This is an important result for the field in itself, which also shines light on possible LOV domain evolution as well as may be further used in optogenetics and biotechnology.

Regarding the results and discussion, the functional experiments are performed and reported with a great attention to detail and are fully convincing. However, I have some comments about the mechanistical part that I think need to be addressed:

1) Paper <https://doi.org/10.1074/jbc.M109.033316> tests some of the same mutations. While the results for the Q123A mutation are similar, results for Q123N are different. Paper <https://doi.org/10.1021/ja1097379> also reports no conformational changes for the Q123N variant of YtvA. I think this needs to be discussed.

2) It is noteworthy that the conformational changes in the mutated AsLOV2 mirror those in WT. However, the light-adapted state structures have undergone radiation damage (as also indicated by the authors) and thus may not faithfully represent the activated LOV domain. Is the chromophore in the NSQ radical form? Could there be additional electron or proton transfer reactions upon radiolysis that are not observed by X-ray crystallography but affect the geometry of the active site?

3) In AsLOV2 structure 2v0u, N414 and D515 are in contact with a glycerol molecule also coordinated by

K489 from another protein molecule. Is it present in the authors' structures? Could this glycerol affect the conformation of N414 and D515? What happens to it in the illuminated state?

4) The authors are not the first to conduct MD simulations of AsLOV2. They cite references 23 and 24 in the introduction, but do not compare their own results against the previous findings. Notably, Q513L was also studied in ref. 23. Both studies highlight the importance of the hydrogen bonds formed by Q513. Are the simulations and mechanisms reported by the authors in agreement with those from refs. 23 and 24?

5) In the lines 406-411 and 521-524, the authors mention electrostatics calculation. However, it is not fully clear how the calculations were performed (software, parameters etc). Fig. S10 legend says "Electrostatic potential acting on the flavin chromophore and residue N414 in the dark-adapted state of wild-type AsLOV2". Presumably, the depicted potential originates from the rest of the protein? For the wild-type protein, fig. S10B shows Q513 in the dark-state conformation, is this correct? I would expect to see the electrostatic forces experienced by N414 resulting from FMN/NSQ and the rest of the protein.

6) Based on simulations and other reasoning, the authors propose that the signal may be transduced via water-mediated hydrogen bonding rearrangements after the protein changes its structure. Could such large scale conformational changes happen in the crystal and lead to rearrangement of N414? Would the crystal conserve its integrity after such a perturbation?

Minor comments:

7) Please include the difference electron density maps for the active site between the dark-adapted and light-adapted crystals in supplementary materials.

8) Please include the traditional protein and chromophore RMSD plots for all of the conducted simulations in supplementary materials.

9) Fig. 5, what is the third C450 conformation in panels c and d? Only two conformations are described in the text.

Reviewer #2 (Remarks to the Author):

This ms is a valuable contribution to the full understanding of signal transmission in the blue-light photosensors of the LOV type. Specifically it explores, by several experimental and computational techniques, the functional role of a glutamine conserved in the majority of LOV domains and generally held responsible for conveying the signal from the chromophore cavity to the surface of the LOV domain. Although the issue is not entirely new, it is here for the first time elegantly and thoroughly explored. I have just minor concerns:

l. 27, l. 58, l. 79 and caption of fig. 1: this glutamine is not strictly conserved, given that several LOV domains do not have it (line 32, line 94, fig. 6)

caption of figure 1: the sequence G.NCRFLQ is not strictly conserved: e.g. in *Pseudomonas putida* it is Y.DCRFLQ and the Q residue is sometimes changed into alanine or leucine

p.3: The following reference should be included: Avila-Pérez, Marcela, et al. "In vivo mutational analysis of YtvA from *Bacillus subtilis*: mechanism of light activation of the general stress response." *Journal of Biological Chemistry* 284.37 (2009): 24958-24964. The paper shows that for the LOV photoreceptors YtvA mutation Q123A does not abolish light regulation in vivo (it is only diminished), while mutation Q123N impairs it.

Fig. S2: although it is true that secondary structures are present in the shown variants, Q123L exhibits smaller light induced conformational changes than wt YF1, actually in line with the smaller light response as reported in fig. 2e. In contrast the light response in fig. 2b is not different from YF1, and so is for NmPAL Q347L. Could the authors comment on this apparent discrepancy?

l. 187: given the results in Avila-Pérez it is not entirely unexpected.

lines 306-307: can this hypothesis be substantiated by data?

Fig. 5: the same simulations as in e and f should be reported for WT protein, in order to support the conclusions at pg. 19

Response to Reviewers

Original comments in italics, responses in red.

Referee #1:

(...)

1) Paper <https://doi.org/10.1074/jbc.M109.033316> tests some of the same mutations. While the results for the Q123A mutation are similar, results for Q123N are different. Paper <https://doi.org/10.1021/ja1097379> also reports no conformational changes for the Q123N variant of YtvA. I think this needs to be discussed.

We appreciate this comment which resonates with remarks by reviewer #2 (see below). In response, we expand the text by a discussion of the photocalorimetric and *in vivo* measurements on the Q123A and Q123N exchanges in *Bacillus subtilis* YtvA (Avila-Pérez *et al.* *J Biol Chem* 2009; Raffelberg *et al.* *J Am Chem Soc* 2011). Indeed, these measurements showed that light-induced conformational and biological responses are absent in YtvA Q123N. These findings contrast with our data on the corresponding YF1 Q123N variant (see Fig. 2b) which reveal attenuated light responses. This discrepancy cannot be resolved at present but may be connected to the different effector modules in the YtvA and YF1 LOV receptors. In closing, we note that the corresponding Q513N exchange in the paradigm AsLOV2 domain did not completely abolish light responses either but merely attenuated them (Nash *et al.* *Biochemistry* 2008).

2) It is noteworthy that the conformational changes in the mutated AsLOV2 mirror those in WT. However, the light-adapted state structures have undergone radiation damage (as also indicated by the authors) and thus may not faithfully represent the activated LOV domain. Is the chromophore in the NSQ radical form? Could there be additional electron or proton transfer reactions upon radiolysis that are not observed by X-ray crystallography but affect the geometry of the active site?

To assess radiation damage during data acquisition, we evaluated the mean and standard deviation, I_{obs} and σ_I , respectively of the integrated diffraction spots across the frames during data collection. As shown below, neither I_{obs} , σ_I , nor their ratio much changed, indicating that overall crystal integrity was maintained during data collection.

That said, we concur with the reviewer that the light-adapted structures experienced radiation damage of its thioadduct, and we had noted this important aspect in the original manuscript. The chemical nature of the radiolysis product(s) of the flavin-cysteiny l thioadduct is not known at present. As described in the manuscript, the chromophore in the light-adapted structures could be modelled by an oxidized flavin in its quinone state, indicating that X-ray radiation did not incur gross structural distortion of the flavin.

Regarding a potential effect of radiation damage on light-induced conformational transitions, we note that the structural differences observed presently between dark-adapted and light-adapted crystals well agree with the previous investigation by Halavaty & Moffat (Biochemistry 2007). Notably, this study also included diffraction experiments at ambient temperature (i.e. non-cryo) under constant blue-light illumination. The resultant structure of light-adapted *AsLOV2* wild-type (PDB 2v1b) showed electron density for the flavin-cysteiny l adduct and N414 reorientation. As our structures of both *AsLOV2* wild-type and Q513L exhibit essentially the same conformational transitions of N414, they likely reflect events genuinely induced by light. We cannot fully rule out additional proton and electron transfer reactions triggered by X-rays, but if they occur, their impact on observable structural differences is likely small.

As for potential X-ray-driven reduction of the flavin to the NSQ state, we note that the NSQ state is predicted to adopt largely planar conformation akin to the quinone state (see, e.g., Kar *et al.* (2021) *WIREs Comput Mol Sci*, <https://doi.org/10.1002/wcms.1541> and references therein). We therefore cannot confidently rule out (fractional) population of the NSQ state. Flavin reduction to the NSQ by the X-rays thus represents an intriguing possibility which we intend to explore in detail in the future. We therefore thank this reviewer for the insightful comment.

3) In *AsLOV2* structure 2v0u, N414 and D515 are in contact with a glycerol molecule also coordinated by K489 from another protein molecule. Is it present in the authors' structures? Could this glycerol affect the conformation of N414 and D515? What happens to it in the illuminated state?

To assess this interesting concept, we inspected the previous *AsLOV2* structures (Halavaty & Moffat, *Biochemistry* 2007, PDB 2v0u [dark], 2v0w [light], 2v1a [room temp., dark], 2v1b [room temp., light]) and compared them to the structures determined presently. Glycerol molecules in the indicated position were modelled in the dark-adapted state of wild-type *AsLOV2* (i.e. in 2v0u and the new structure 7pgx). In all other past and present structures (2v0w, 2v1a, 2v1b, 7pgy, 7pgz, and 7ph0), no glycerol was identified in this location, and the electron density in this region was better described by water molecules. However, we caution that these observations do not strictly prove the absence of glycerol in this location, because structurally disordered glycerol may still be present.

Taken together, the presence of the mentioned glycerol molecule is therefore consistent across the previous and current structures for wild-type AsLOV2 (2v0u and 7pgx have it, but 2v0w and 7pgy do not). Qualitatively very similar light-induced structural changes manifest in the presence of a structurally resolvable glycerol (2v0u vs. 2v0w, and 7pgx vs. 7pgy) and in its absence (2v1a vs. 2v1b, and 7pgz vs. 7ph0). Although we cannot strictly rule out a glycerol contribution to signal transduction, based on the above evidence, we deem it unlikely.

4) *The authors are not the first to conduct MD simulations of AsLOV2. They cite references 23 and 24 in the introduction, but do not compare their own results against the previous findings. Notably, Q513L was also studied in ref. 23. Both studies highlight the importance of the hydrogen bonds formed by Q513. Are the simulations and mechanisms reported by the authors in agreement with those from refs. 23 and 24?*

The pioneering simulations by Freddolino *et al.* (*Biophys J* 2006, reference no. 23) revealed the importance of light-induced hydrogen-bonding changes, in particular of the flavin and Q513. These aspects are in general agreement with our simulations. Further, the *Biophys J* work proposed that signal transduction in phototropin LOV2 domains mainly involves structural changes in the H β -I β loop, which contrasts with our findings that point at the β -sheet interface and the A' α helix (see Fig. 5 and Suppl. Fig. S7). However, the same principal authors later refined their simulations (Freddolino *et al.* *Photochem Photobiol Sci* 2013). The updated model again emphasizes hydrogen-bonding interactions but now pinpoints the β -sheet:A' α interface as a crucial conduit for signaling, which agrees with our studies. Interestingly, in simulations on the light-adapted state the authors observed a water molecule in hydrogen-bonding contact to the (protonated) N5 atom, even with Q513 present. In addition, the authors also interrogated the Q513L mutation and found that small structural changes still occur, but large-scale effects are abolished due to the inability of the leucine sidechain to enter hydrogen bonds. Taken together, the principal signaling trajectory for AsLOV2 wild-type proposed by Freddolino *et al.* closely matches our experimental observations and simulations. By contrast, the previous simulations posited that Q513L is incapable of efficient downstream signaling. This conclusion however is incompatible with the experimental data (see Figs. 4c, 4d, 5d and Suppl. Fig. S7d). Our simulations now implicate water influx as a mechanism to relay hydrogen-bonding signals in the absence of the glutamine.

We greatly appreciate this reviewer comment as it prompted us to revisit the earlier literature. Doing so, we realized that we inadvertently left out the important 2013 work by Freddolino *et al.* We now rectify this omission and refer to these two simulation studies in more detail in the Results and the Discussion sections.

The recent work by Iuliano *et al.* (*ACS Chem Biol* 2020, originally cited as reference 24, now no. 25) reports simulations on AsLOV2 which suggest that reorientation of Q513 is aided by transient rearrangements of the asparagines 482 and 492 that coordinate the flavin cofactor (see Fig. 5). A potential contribution of these two residues would be compatible in principle with our signaling model in the presence of the glutamine (see Suppl. Fig. S16). Intriguingly, Iuliano *et al.* propose that in the light-adapted/signaling state, Q513 hydrogen-bonds to the flavin atom O4 rather than to the newly protonated N5 atom, as conventionally assumed and suggested by crystallographic evidence (see, e.g., Fig. 5). Based on our experimental data and simulations, we can however neither affirm nor contradict this proposition with sufficient confidence.

Our original manuscript already covered the Iuliano *et al.* work but we now expand the Discussion section accordingly.

5) *In the lines 406-411 and 521-524, the authors mention electrostatics calculation. However, it is not fully clear how the calculations were performed (software, parameters etc). Fig. S10 legend*

says "Electrostatic potential acting on the flavin chromophore and residue N414 in the dark-adapted state of wild-type AsLOV2". Presumably, the depicted potential originates from the rest of the protein? For the wild-type protein, fig. S10B shows Q513 in the dark-state conformation, is this correct? I would expect to see the electrostatic forces experienced by N414 resulting from FMN/NSQ and the rest of the protein.

We thank the reviewer for pointing out the missing description of the electrostatics calculations. We expand the corresponding section in the Methods part, now entitled 'Electrostatics and molecular simulations'.

As this reviewer notes, the electrostatic potential experienced by the flavin chromophore and residue N414 originates from the protein environment. It was obtained from two separate calculations but is shown together in one figure. To better convey these aspects, we rephrase the legend to Suppl. Fig. S11 (formerly Suppl. Fig. S10) and expand the Methods chapter.

Panel b of Suppl. Fig. S11 indeed shows Q513 in its dark-state conformation, i.e. with its sidechain amide NH₂ group oriented towards the flavin chromophore. We chose this approach to investigate the effect of thioadduct formation and concomitant N5 protonation on electrostatics in the absence of other hydrogen-bonding changes. We now describe these points in the Methods section.

6) *Based on simulations and other reasoning, the authors propose that the signal may be transduced via water-mediated hydrogen bonding rearrangements after the protein changes its structure. Could such large scale conformational changes happen in the crystal and lead to rearrangement of N414? Would the crystal conserve its integrity after such a perturbation?*

As correctly noted by the reviewer and as stated in the manuscript, the crystal lattice may limit the scope and scale of conformational changes compared to in solution. This aspect is inherent to freeze-trapping strategies and well exemplified by the terminal A'α and Jα helices which undergo light-induced unfolding in solution but not in the crystal.

That said, our structures do show N414 reorientation in both AsLOV2 wild-type and Q513L (see Fig. 5 and Suppl. Fig. S8) and thereby demonstrate that conformational rearrangements principally occur in the crystal. The proposed signaling mechanism for Q513L mainly differs from that for wild type in that it relies on the influx of water molecules. As N414 is directly exposed to the solvent, we expect that small-scale structural fluctuations likely suffice to allow water entry.

Minor comments:

7) *Please include the difference electron density maps for the active site between the dark-adapted and light-adapted crystals in supplementary materials.*

We calculated $F_{\text{light}} - F_{\text{dark}}$ difference density maps for AsLOV2 wild-type and Q513L. In case of the Q513L variant, the best-resolved dark and light datasets, used for structure refinement (7pgz and 7ph0), differed in the *b* dimension of the unit cell by 1.5 Å (see Suppl. Table S2). For map calculation, we therefore selected a different light dataset at a resolution of 1.24 Å in which this difference only amounted to 0.85 Å.

We include the maps as Suppl. Fig. S7, provide the MTZ files underlying the difference maps as Supplementary Data, and expand the Methods section accordingly.

8) *Please include the traditional protein and chromophore RMSD plots for all of the conducted simulations in supplementary materials.*

We provide the regular RMSD values along the MD simulations as Suppl. Fig. S12.

9) Fig. 5, what is the third C450 conformation in panels c and d? Only two conformations are described in the text.

In Fig. 5, we assign two principal orientations of residue C450, conformation 'a' pointing to the left and conformation 'b' pointing to the right (in the view chosen for Fig. 5). In the structures of AsLOV2 wild-type light-adapted (panel b), Q513L dark-adapted (panel c) and light-adapted (panel d), the conformation 'b' of C450 splits into two subpopulations with somewhat different chi1 angles. We now note this fact in the legend to Fig. 5.

Referee #2:

(...)

l. 27, l. 58, l. 79 and caption of fig. 1: this glutamine is not strictly conserved, given that several LOV domains do not have it (line 32, line 94, fig. 6)

Agreed. We replace the word 'strictly' by 'strongly' or 'highly'.

caption of figure 1: the sequence G.NCRFLQ is not strictly conserved: e.g. in Pseudomonas putida it is Y.DCRFLQ and the Q residue is sometimes changed into alanine or leucine

Again, this reviewer is correct in that the conservation is not absolute. To reflect this aspect, we rephrase the sentence to read 'strongly conserved'. The assignment of these residues is based on Crosson's and Moffat's 2003 review which we cite in the figure legend.

p.3: The following reference should be included: Avila-Pérez, Marcela, et al. "In vivo mutational analysis of YtvA from Bacillus subtilis: mechanism of light activation of the general stress response." Journal of Biological Chemistry 284.37 (2009): 24958-24964. The paper shows that for the LOV photoreceptors YtvA mutation Q123A does not abolish light regulation in vivo (it is only diminished), while mutation Q123N impairs it.

We are grateful for this comment which is similar to remarks by reviewer #1 (see point 1). As detailed above, we now include a discussion of the important YtvA work in our manuscript. We also highlight the divergent findings for the Q123N exchange, see above.

Fig. S2: although it is true that secondary structures are present in the shown variants, Q123L exhibits smaller light induced conformational changes than wt YF1, actually in line with the smaller light response as reported in fig. 2e. In contrast the light response in fig. 2b is not different from YF1, and so is for NmPAL Q347L. Could the authors comment on this apparent discrepancy?

This is an important point. When studied at the level of the purified proteins, the replacement of glutamine by leucine (Q123L in YF1, Q347L in PAL, and Q513L in AsLOV2) invariably diminished the amplitude of light-induced downstream signaling responses (see Figs. 2e, 3e, 3f, 4b, and 4d). However, as correctly noted by this reviewer, the bacterial reporter-gene assays for both YF1 and PAL showed only small effects of the exchange of glutamine by leucine (see Figs. 2b and 3b).

These findings can be rationalized by noting that the bacterial assays have a limited dynamic range of signals that can be resolved. Moreover, these assays only indirectly report on the molecular light-dependent receptor activity (see Figs. 2a and 3a). The reporter-gene output is thus expected to experience saturation and thresholding effects. For instance, in case of YF1 the enzymatic assay (Fig. 2e) clearly showed reduced light responses for Q123L; however, inside the bacteria this reduced response still sufficed to down-regulate reporter output to nearly the same extent as the original YF1 (Fig. 2b).

l. 187: given the results in Avila-Pérez it is not entirely unexpected.

We replace 'unexpected' by 'striking'.

lines 306-307: can this hypothesis be substantiated by data?

We neither have data to substantiate this hypothesis, nor can we devise a suitable experiment to obtain such data. That is because the proposed contribution to the CD signal by flavin photoreduction will only manifest when the chromophore is embedded within the chiral surroundings (with respect to polarizability) of the chromophore-binding pocket. Hence, it cannot be disentangled from CD signal changes caused by secondary structure transitions within the protein scaffold.

We originally stated this hypothesis to provide a possible (and hopefully plausible) explanation for the observed spectral signals. In addition, we think it is important to make readers aware that at least in principle photochemical events within the chromophore can contribute to the observable CD signal.

That said, we realize that our original statement could be construed as if the contribution of flavin photochemistry was a proven fact. To prevent this (unintended) interpretation, we now explicitly state that the origin of the signal change observed in Fig. 4f is unknown.

Fig. 5: the same simulations as in e and f should be reported for WT protein, in order to support the conclusions at pg. 19

Such simulations were already provided in Suppl. Fig. S13 (formerly Suppl. Fig. S11), which we now point out in the legend of Fig. 5a.

REVIEWER COMMENTS

Reviewer #1 (Remarks to the Author):

In the revised version, the authors have addressed my comments and resolved most of them. As I wrote earlier, the manuscript presents an important advance for the field and contains interesting data. I would recommend publication after the relatively minor issues listed below are resolved.

Keeping the same enumeration as in the original review,

5) The point of electrostatics calculations as presented is not clear to me. After QM/MM geometry optimization, the structure of the protein is presumably almost the same. On the other hand, in vivo/in vitro it is possible that the protein relaxes and/or fluctuates differently in the dark- and light-adapted states, and the electrostatic forces acting on N414 are different on average, but not in this specific X-ray conformation used in calculations.

Also, the claim that "changes in the electrostatic potential [are] largely confined to the chromophore itself" is not addressed by this calculation at all. To claim this, the authors would need to calculate the electrostatic forces originating from the flavin charges that act on the protein atoms surrounding the flavin and show that the changes are indeed not significant beyond the immediate vicinity. Yet, anyway, electrostatic forces decay very fast with distance, and are screened by other chemical groups.

The point of calculating the electrostatic potential experienced by the flavin is also not clear. If the protein model is essentially the same, then the potential is not expected to change. Also, when calculating the potential around the flavin in the light-adapted state, were the cysteine atoms part of the flavin moiety? In any case, electrostatic potential calculated in such a way would include intramolecular electrostatics, which is likely meaningless without taking into account all other energetic contributions from covalent bonding between the flavin and the protein.

7) For the difference electron density maps (Fig. S8), I do not get the explanation for the dominance of the negative densities. The maps also look unexpectedly noisy. Usually, one would like to see clear paired densities for each residue that changes its conformation (as observed for the cysteine). The maps in the paper by Halavaty & Moffat (Biochemistry 2007, Fig. S3) look much better in this regard, despite the overall lower resolution. I cannot pinpoint the possible reason for the noisier maps in the manuscript under review, only perhaps that the maps should be calculated using the same resolution cut-off for both datasets: if it was done so, it is not clear from the methods description. And/or the authors should use a higher contouring level or try a different software for calculating the maps.

Finally, page 9/line 363, the meaning of "As presently" is not clear.

Reviewer #2 (Remarks to the Author):

All comments and questions have been properly addressed. The ms is ready for publication

Response to Reviewers

Original comments in italics, responses in red.

Referee #1:

(...)

5) The point of electrostatics calculations as presented is not clear to me. After QM/MM geometry optimization, the structure of the protein is presumably almost the same. On the other hand, in vivo/in vitro it is possible that the protein relaxes and/or fluctuates differently in the dark- and light-adapted states, and the electrostatic forces acting on N414 are different on average, but not in this specific X-ray conformation used in calculations.

Also, the claim that "changes in the electrostatic potential [are] largely confined to the chromophore itself" is not addressed by this calculation at all. To claim this, the authors would need to calculate the electrostatic forces originating from the flavin charges that act on the protein atoms surrounding the flavin and show that the changes are indeed not significant beyond the immediate vicinity. Yet, anyway, electrostatic forces decay very fast with distance, and are screened by other chemical groups.

The point of calculating the electrostatic potential experienced by the flavin is also not clear. If the protein model is essentially the same, then the potential is not expected to change. Also, when calculating the potential around the flavin in the light-adapted state, were the cysteine atoms part of the flavin moiety? In any case, electrostatic potential calculated in such a way would include intramolecular electrostatics, which is likely meaningless without taking into account all other energetic contributions from covalent bonding between the flavin and the protein.

We fully agree with this reviewer's principal notion that electrostatic forces decay fast with distance. To test and possibly substantiate this sentiment, we evaluated in our manuscript the impact of altered electrostatics caused by thioadduct formation and N5 protonation, but in the absence of other structural changes. As the reviewer correctly surmises, the underlying QM/MM-optimized models for the dark-adapted and light-adapted states used to this end are structurally highly similar. We further agree that in reality (i.e. 'in vivo/in vitro') these two states differ more extensively in their structural dynamics, as not least evidenced by the light-promoted A' α /J α helix unfolding. However, we reiterate that the computational approach was deliberately chosen to hypothetically probe how altered electrostatics would play out in isolation, i.e. without downstream changes in structural dynamics, and whether signal transduction all the way to N414 could thus be achieved.

That said, we realize that our presentation of the simulations left wanting and was potentially misleading. Following this reviewer's advice, we expanded the electrostatics calculations in Suppl. Fig. S11 to also encompass residues Q513/L513, N482, and N492, which have been implicated in AsLOV2 signal transduction (see Introduction and Discussion sections). The new evaluation shows that Q513/L513 experiences subtly altered electrostatics due to protonation of the directly adjacent flavin N5, but in the absence of other structural rearrangements, the effect tapers off fast with distance and does not reach N414, N482, or N492. From these observations, we conclude that altered electrostatics originating at the flavin chromophore do not propagate sufficiently far to affect N414 directly. Rather, other structural changes, especially hydrogen-bonding changes, are required to relay light signals. Again, we note that this result is in line with this reviewer's and our own expectations.

Finally, we agree that showing the potential surface for the flavin chromophore was misleading. That is because meaningful simulations would need to account for light-induced changes of AsLOV2 structure and dynamics, as correctly remarked by the reviewer. Moreover, our simulations sought to probe the effect of thioadduct formation emanating from the flavin to the surroundings, and not *vice versa*. We have thus removed from Suppl. Fig. S11 the electrostatic potential map of the flavin and added maps for residues N482, N492, and Q513/L513. Further, we have revised the figure legend and manuscript to incorporate the changes laid out above.

7) For the difference electron density maps (Fig. S8), I do not get the explanation for the dominance of the negative densities. The maps also look unexpectedly noisy. Usually, one would like to see clear paired densities for each residue that changes its conformation (as observed for the cysteine). The maps in the paper by Halavaty & Moffat (Biochemistry 2007, Fig. S3) look much better in this regard, despite the overall lower resolution. I cannot pinpoint the possible reason for the noisier maps in the manuscript under review, only perhaps that the maps should be calculated using the same resolution cut-off for both datasets: if it was done so, it is not clear from the methods description. And/or the authors should use a higher contouring level or try a different software for calculating the maps.

We greatly appreciate this astute comment. When we carefully examined the joint dark/light datasets that underpinned the difference density maps in Suppl. Fig. S8, we noticed a problem with data scaling. Inadvertently, the “dark” data thus acquired a higher weight than the “light” data. As a consequence, the original $F_{\text{light}}-F_{\text{dark}}$ difference maps were dominated by the “dark” data and showed more negative (red) signals than positive (green) ones.

We have now rescaled the data and recalculated the maps (see updated Suppl. Fig. S8). The new $F_{\text{light}}-F_{\text{dark}}$ maps show about the same amounts of positive and negative difference density (as correctly expected by this reviewer). Moreover, the updated maps have less noise which to significant extent had originated from the mis-scaling. Beyond updating Suppl. Fig. S8, we also include as Suppl. Data 1 and 2 updated MTZ files for difference density calculation.

Further, this reviewer refers to Suppl. Fig. S3 in the Halavaty & Moffat work which shows a close-up view of the FMN chromophore. Thus, only a small portion of the difference density map, contoured at $\pm 3.0 \sigma$, is visible. To better compare to our data, we calculated a $F_{\text{light}}-F_{\text{dark}}$ difference density map based on the PDB entries 2v0u (dark) and 2v0w (light). At a contour level of $\pm 4.0 \sigma$ (as chosen by Halavaty & Moffat in their Fig. 4), the map has comparable signal-to-noise ratio as our data (also at $\pm 4.0 \sigma$).

Finally, we refer to our (updated) Suppl. Fig. S8a. The difference density signals are spatially concentrated on the right half of the molecule whereas at a level of $\pm 4.0 \sigma$ no signal is detected in the left half of the molecule. This observation strongly argues that the signals in the maps are not spurious but genuinely reflect differences between the “dark” and “light” datasets.

Finally, page 9/line 363, the meaning of "As presently" is not clear.

We have rephrased this sentence. (“As in the present study, ...”).

REVIEWERS' COMMENTS

Reviewer #1 (Remarks to the Author):

The authors have addressed all of the comments and the manuscript is ready for publication.

Response to Reviewers and Editor

Original comments in italics, responses in red.

Referee #1:

The authors have addressed all of the comments and the manuscript is ready for publication.

We thank the reviewer for his/her efforts and for approving of our work.

Editor:

We have addressed all points raised in the authors checklist and in the marked-up reporting summary PDF.